# Few-Shot Idea Auto-Generation: Reasoning Over Idea Representations to Predict New Research Ideas

## Abstract

Large language models have demonstrated powerful reasoning capabilities on user-provided contexts, inspiring researchers to explore their potential for automated research. A critical component of research is idea generation—identifying novel contributions, advantages, and distinctions from existing work. However, we show that naively prompting pre-trained LLMs to generate research ideas produces largely meaningless results. We introduce a novel task: few-shot idea auto-generation, where models generate research ideas based on a small set of existing papers. Our key insight is that meaningful ideas typically build upon prior work rather than emerging from scratch—for instance, adapting solutions from one domain to address similar challenges in another, often combined with novel algorithmic approaches. To enable effective few-shot idea generation, we address three fundamental challenges: (1) How can we effectively represent the core ideas of existing papers? (2) How can we generate practical, implementable ideas while filtering out infeasible ones? (3) How can we validate the generated ideas effectively? Our contributions are threefold. First, we develop an idea representation method that effectively captures papers' core contributions through *multi-agent extraction with synopsis and procedural profiling*. Second, we design an LLM-agent-based generation framework that *performs cross-pollination via systematic gap-bridging between paper pairs*. Third, we propose an evaluation methodology using *semantic similarity analysis with recency-weighted novelty scoring* and construct a benchmark for *few-shot idea generation across 3,353 papers from 8 computer science domains*.

## 1 Introduction

The emergence of Agentic AI marks a transformative shift in scientific research, where autonomous systems can execute complete research workflows—from hypothesis generation to manuscript preparation—with minimal human intervention (Gridach et al., 2025; Gao et al., 2024; Lu et al., 2024). Unlike traditional task-specific AI tools, these systems demonstrate sophisticated reasoning, planning, and decision-making capabilities that enable independent scientific inquiry. Recent implementations showcase this evolution through *AI Scientists*—autonomous systems that generate hypotheses, conduct experiments, and write papers. Notable examples include *The AI Scientist* (Lu et al., 2024) and *AI Scientist-v2* (Yamada et al., 2025), which produce results competitive with human researchers in machine learning domains. Collaborative frameworks have also emerged: *Chain-of-Ideas* (Li et al., 2024) and *SciAgents* (Ghafarollahi & Buehler, 2024b) simulate research teams for collective problem-solving, while industry initiatives like Google DeepMind's *AI Co-Scientist* accelerate biomedical research through domain-specialized multi-agent systems. Despite achieving notable successes—including papers that pass peer review—current systems face significant limitations. They prioritize end-to-end automation over deep engagement with individual research stages, potentially missing critical nuances. Studies reveal that while AI agents can generate novel ideas, they often lack feasibility (Si et al., 2025; Weng et al., 2025), highlighting the gap between creative ideation and practical implementation.

We focus on the foundational stage of AI scientist, Idea Generation, where novel research concepts are conceived and formulated. As illustrated in Figure 1, our approach takes existing papers and

their conceptual representations as input to a few-shot idea generator, producing multiple research ideas. **These automatically generated ideas serve only as inspirational prompts for human researchers, who remain responsible for critical evaluation and creative decision-making.** We evaluate the system's impact on future studies using three metrics: similarity between generated ideas and subsequently published papers, uniqueness ratios of matched papers, and novelty scores.

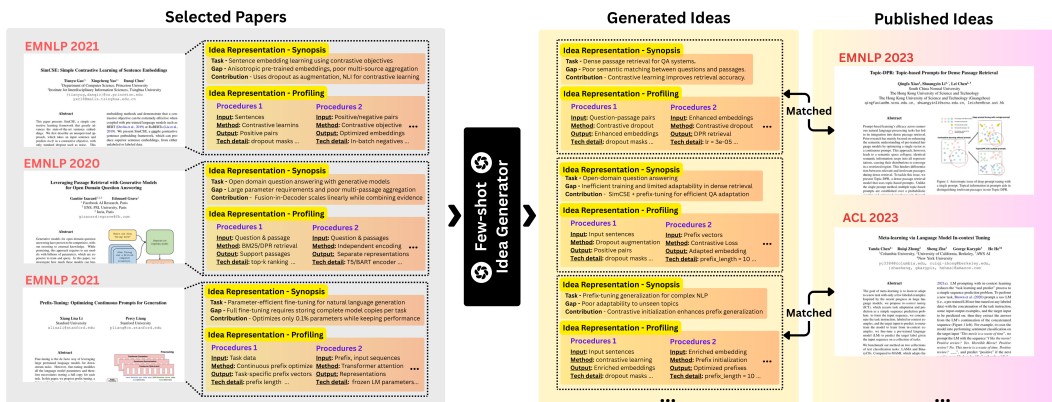

Figure 1: Example of our few-shot idea auto-generation framework. Using three foundational studies, SimCSE (Gao et al., 2021), Fusion-in-Decoder(Izacard & Grave, 2021), and Prefix-Tuning (Li & Liang, 2021), published in 2021-2022, we employ idea representation to encode parent papers, then use a full representation-based generator to create new research ideas based on every two papers, which are subsequently matched the papers published later with high similarity, Topic-DRP (Xiao et al., 2023) and Meta-learning (Chen et al., 2022).

## 2 MOTIVATION AND INSIGHTS

Our investigation is motivated by several key insights about the nature of scientific innovation and current limitations in automated research idea generation.

**(1) Problem-driven vs. Gap-driven**   Current research idea generation methods are predominantly problem-driven. Approaches like CoI (Li et al., 2024), SciAgents (Ghafarollahi & Buehler, 2024b), and AI Scientist (Lu et al., 2024) follow a generation-review loop that depends on subjective review agents to assess novelty and feasibility. Two fundamental approaches exist: (a) identifying worthy problems through intensive literature review, or (b) finding gaps between specific studies. Problem-driven approaches require resource-intensive analysis, rely on opaque LLM judgments, and provide no transparency about idea emergence. In contrast, bridging gaps between specific studies is more traceable and resouce saving, as illustrated in Figure 2. This gap-driven approach offers explicit control over what is transferred from each study to construct new ideas. We therefore adopt this systematic approach of identifying and addressing gaps through targeted cross-pollination.

**(2) Importance of Idea Representation**   In terms of study understanding, the general summaries of paper sections used in current approaches (as seen in CoI (Li et al., 2024), SciAgents (Ghafarollahi & Buehler, 2024b), and AI Scientist (Lu et al., 2024)) lack sufficient detail for identifying meaningful research gaps—a limitation we demonstrate in our experiments. This presents the serious challenge on how to effectively analyze prior research to understand their core contributions and develop appropriate representations of individual research insights that preserve critical details.

**(3) Empirical Evaluation**   Existing agent-related studies typically score machine-generated research ideas using AI-based rubrics for novelty, feasibility, significance, clarity, and effectiveness (Li et al., 2024; Ghafarollahi & Buehler, 2024b; Lu et al., 2024; Shahid et al., 2025). Other novelty measures remain proxy-based, using edge-factor scores (Packalen, 2018), co-citation z-scores (Uzzi et al., 2013), or word-embedding distances (Shibayama et al., 2021). These methods either lack prospective validation on AI-generated ideas or rarely trace idea provenance to specific prior works, limiting their ability to predict real-world impact. These limitations motivate developing

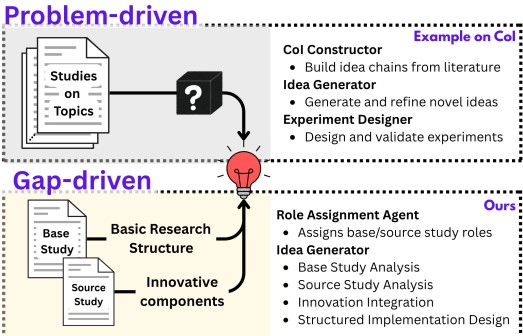

Figure 2: **Demonstration of how problem-driven and gap-driven methods approach idea generation**: Problem-driven approaches, such as CoI, begin with a literature review on a given topic and build a Chain of Ideas by tracing forward and backward references. An agent then generates ideas based on this chain, but the structure of the final idea is not predictable until after generation. In contrast, gap-driven approaches require only two papers, making the process more controllable—the user explicitly knows what is being transferred from each study to construct the new idea.

literature-grounded frameworks that map generated ideas onto the scholarly landscape while providing tunable, transparent scoring mechanisms.

## 3 METHODOLOGY

The research methodology comprises three core components: idea representation, idea generation, and idea evaluation. Figure 1 illustrates the complete pipeline, focusing on how idea representations are extracted from selected papers and integrated to synthesize new research concepts, while also presenting the full evaluation process through matching with subsequently published work.

### 3.1 IDEA REPRESENTATION

Idea representation employs specialized extraction functions—namely, $f_{task}$, $f_{gaps}$, $f_{contrib}$, and $f_{proc}$—built on GPT-4o-mini (OpenAI, 2024), where each agent handles a specific aspect of information extraction. This modular approach enables focused, expert-level processing of different paper components.

Given a paper denoted as $\mathcal{P}$, we split it into different sections to obtain $\mathcal{P} = \{\mathcal{I}, \mathcal{M}, \ldots\}$, where $\mathcal{I}$ is the introduction section and $\mathcal{M}$ represents the method section. Our goal is to produce a comprehensive, structured representation from $\mathcal{P}$ that integrates synopsis and procedural profiling: $\mathcal{R} = \{\mathcal{T}, \mathcal{G}, \mathcal{C}\} \cup \{\mathcal{S}_{proc}\}$, where $\mathcal{R}$ denotes the complete structured paper representation, $\mathcal{T}$ represents the core research task, $\mathcal{G}$ contains identified research gaps, $\mathcal{C}$ encompasses the paper's contributions, and $\mathcal{S}_{proc}$ captures detailed procedural methodologies. Examples of this representation are shown in figure 1.

#### 3.1.1 IDEA REPRESENTATION - SYNOPSIS

The synopsis extraction combines task, gap, and contribution identification: $\mathcal{T} = f_{task}(\mathcal{I})$, $\mathcal{G} = f_{gaps}(\mathcal{I}) = \{g_1, g_2, \ldots, g_n\}$, and $\mathcal{C} = f_{contrib}(\mathcal{I}, \mathcal{G}) = \{(g_i, c_i) \mid g_i \in \mathcal{G}, c_i \in \mathcal{C}\}$. Function $f_{task}$ extracts action-oriented task descriptions, $f_{gaps}$ identifies 2-5 technical limitations, and $f_{contrib}$ establishes explicit gap-contribution mappings, ensuring traceability between problems and solutions.

#### 3.1.2 IDEA REPRESENTATION - PROFILING

The profiling process extracts detailed procedural information $\mathcal{S}_{proc}$ from the methodology section $\mathcal{M}$. The procedural extraction function maps methodology content to structured input-method-output-details quadruplets: $\mathcal{S}_{proc} = f_{profile}(\mathcal{M}) = \{\langle I_k, M_k, O_k, D_k \rangle \mid k = 1, \ldots, K\}$, where $\mathcal{M}$ represents the methodology section, $K$ is the total number of procedural steps, $I_k$ represents input components, $M_k$ denotes methodological processes, $O_k$ captures output specifications, and $D_k$

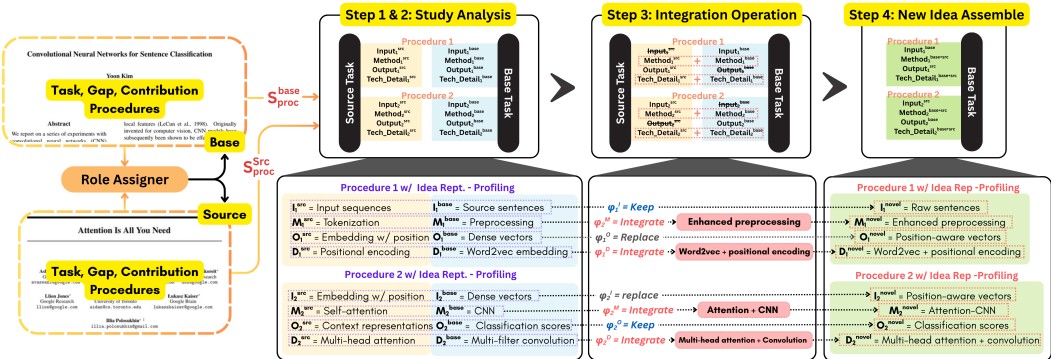

Figure 3: **Cross-pollination idea generation example through role assignment and integration.** Two well-known studies are first assigned roles: *Convolutional Neural Networks for Sentence Classification* (Kim, 2014) serves as the *base study* providing the problem domain of text classification with a methodological gap in capturing long-range dependencies, while *Attention Is All You Need* (Vaswani et al., 2017) serves as the *source study* contributing the self-attention innovation. Procedural quadruplets from both studies are then systematically integrated using operations to generate a novel methodology that combines CNN classification with attention mechanisms.

contains technical details including parameters, algorithms, and implementation tools for the $k$-th procedural step.

## 3.2 IDEA GENERATION

The role assignment determines which paper serves as the base study (task anchor) and which provides the innovation source for cross-pollination. The role assigner analyzes synopsis components from both papers:

$$\mathcal{R}^* = G_{A_r}(\{\mathcal{T}_1, \mathcal{G}_1, \mathcal{C}_1\}, \{\mathcal{T}_2, \mathcal{G}_2, \mathcal{C}_2\}) \rightarrow [\mathcal{R}^{\text{base}}, \mathcal{R}^{\text{src}}] \tag{1}$$

where $G_{A_r}$ evaluates problem clarity, innovation strength, and transferability using synopsis components $\{\mathcal{T}_i, \mathcal{G}_i, \mathcal{C}_i\}$. The assignment considers problem clarity through task definition quality in $\mathcal{T}$, innovation potential by analyzing contribution novelty in $\mathcal{C}$, and transferability by matching gaps $\mathcal{G}$ with contributions $\mathcal{C}$ across papers. The output assigns $\mathcal{P}_{\text{base}}$ as the base study and $\mathcal{P}_{\text{src}}$ as the innovation source.

### 3.2.1 IDEA GENERATION

Idea Generation is implemented using a Chain-of-Thought (CoT) agent that systematically walks through each step, ensuring logical coherence and maintaining explicit connections between the base study's limitations and the proposed innovations. An example with the full generation process is illustrated in Figure 3. This agent outputs a structured research proposal with clear implementation steps and expected improvements over the base study. The overall process is formalized as: $\mathcal{R}^{\text{novel}} = G_{A_n}(\mathcal{R}^{\text{base}}, \mathcal{R}^{\text{src}})$, where $\mathcal{R}_{\text{base}}$ represents the structured representation of the base study, $\mathcal{R}_{\text{src}}$ denotes the structured representation of the innovation source paper, and $G_{A_n}$ is the cross-pollination agent that produces a novel research idea $\mathcal{I}_{\text{novel}}$.

**Step 1: Base study analysis** The foundation analysis extracts the complete structured representation $\mathcal{R}^{\text{base}} = \{\mathcal{T}^{\text{base}}, \mathcal{G}^{\text{base}}, \mathcal{C}^{\text{base}}\} \cup \{\mathcal{S}^{\text{base}}_{\text{proc}}\}$ from the base study, where $\mathcal{S}^{\text{base}}_{\text{proc}} = \{\langle I_i, M_i, O_i, D_i \rangle \mid i = 1, \ldots, K_{\text{base}}\}$ captures the procedural quadruplets.

**Step 2: Source study analysis** For the source paper, the agent analyzes the structured representation $\mathcal{R}^{\text{src}} = \{\mathcal{T}^{\text{src}}, \mathcal{G}^{\text{src}}, \mathcal{C}^{\text{src}}\} \cup \{\mathcal{S}^{\text{src}}_{\text{proc}}\}$, focusing on the procedural quadruplets $\mathcal{S}^{\text{src}}_{\text{proc}} = \{\langle I_j, M_j, O_j, D_j \rangle \mid j = 1, \ldots, K_{\text{src}}\}$ to identify transferable innovations.

**Step 3: Integration operation** This adaptation process ensures compatibility between the source innovation and base study methodology through systematic quadruplet composition. Given paired

quadruplets from base and source studies, the agent selects one operation for each component from $\psi \in \{\text{integrate}, \text{replace}, \text{keep}, \text{remove}\}$.

The composition of new quadruplets is formalized as:

$$\langle I_k^{\text{novel}}, M_k^{\text{novel}}, O_k^{\text{novel}}, D_k^{\text{novel}} \rangle = \Theta\Big( \langle I_i^{\text{base}}, M_i^{\text{base}}, O_i^{\text{base}}, D_i^{\text{base}} \rangle,$$
$$\langle I_j^{\text{src}}, M_j^{\text{src}}, O_j^{\text{src}}, D_j^{\text{src}} \rangle, \langle \psi^I, \psi^M, \psi^O, \psi^D \rangle \Big), \tag{2}$$

where $\Theta$ is the composition operator and $\langle \psi^I, \psi^M, \psi^O, \psi^D \rangle$ specifies the operation applied to each component. The integration process identifies correspondences between base study gaps $\mathcal{G}_{\text{base}}$ and source contributions $\mathcal{C}_{\text{src}}$ to guide operation selection. Operation $integrate$ combines complementary components from both quadruplets, $replace$ substitutes base components with source innovations, $keep$ preserves base methodology unchanged, and $remove$ eliminates incompatible or redundant components.

**Step 4: New idea assemble** The generated idea is constructed by systematically applying the composition operator across all procedural steps, where each step involves component-wise operation selection:

$$\mathcal{R}_{\text{proc}}^{\text{novel}} = \Big\{ \Theta(\langle I_i^{\text{base}}, M_i^{\text{base}}, O_i^{\text{base}}, D_i^{\text{base}} \rangle,$$
$$\langle I_j^{\text{src}}, M_j^{\text{src}}, O_j^{\text{src}}, D_j^{\text{src}} \rangle, \Psi_k) \mid k = 1, \ldots, K_{\text{novel}} \Big\}, \tag{3}$$

where $\Psi_k = \langle \psi_k^I, \psi_k^M, \psi_k^O, \psi_k^D \rangle$ represents the operation vector for the $k$-th procedural step, with each component-specific operation ensuring coherent methodology construction through deliberate composition choices.

### 3.3 IDEA EVALUATION

The evaluation methodology assesses three key aspects: semantic similarity between input papers, the uniqueness ratio of matched papers, and the novelty of generated research ideas.

#### 3.3.1 PAPER SIMILARITY

The paper similarity assessment follows a three-step process. First, the generated novel idea $\mathcal{I}\text{novel}$ is distilled into a searchable representation $\mathcal{Q}$ that extracts task with core methodological concepts while removing specific model names and focusing on broad algorithmic categories. Second, academic papers are retrieved using semantic search on the query $\mathcal{Q}$, yielding a candidate set $\mathcal{P}\text{retrieved}$, with source paper exclusion applied: $\mathcal{P}\text{filtered} = \mathcal{P}\text{retrieved} \setminus \{\mathcal{P}\text{base}, \mathcal{P}\text{src}\}$ to ensure that the originating papers from the cross-pollination process do not bias the evaluation. Finally, embedding-based similarity is computed between the novel idea and each retrieved paper using cosine similarity:

$$\sigma(\mathcal{Q}, p_j) = \frac{\mathcal{E}(\mathcal{Q}) \cdot \mathcal{E}(p_j)}{|\mathcal{E}(\mathcal{Q})| \cdot |\mathcal{E}(p_j)|} \tag{4}$$

where $\mathcal{E}(\cdot)$ denotes the text embedding function, $\mathcal{Q}$ represents the distilled novel research idea, and $p_j \in \mathcal{P}_{\text{filtered}}$ denotes individual papers from the filtered retrieval set.

#### 3.3.2 UNIQUE PAPER RATIO

The unique paper ratio quantifies the diversity of research approaches within the papers that exhibit the highest similarity to the generated idea, assessing the methodological landscape of the most relevant existing work. From the filtered paper collection $\mathcal{P}_{\text{filtered}}$, we identify papers with the highest similarity score to the distilled searchable representation $\mathcal{Q}$, forming the highest-similarity subset $\mathcal{P}_{\text{max}} = \{p_j \in \mathcal{P}_{\text{filtered}} \mid \sigma(\mathcal{Q}, p_j) = \max_{p \in \mathcal{P}_{\text{filtered}}} \sigma(\mathcal{Q}, p)\}$. The unique paper ratio is then computed as:

$$\text{U.Ratio} = \frac{|\mathcal{P}_{\text{unique}}|}{|\mathcal{P}_{\text{max}}|} \tag{5}$$

where $\mathcal{P}_{\text{unique}} \subseteq \mathcal{P}_{\text{max}}$ represents the subset of papers that employ distinct methodological approaches, determined through clustering analysis of their procedural components. A higher UPR indicates greater methodological diversity among the most similar papers, suggesting that the generated idea operates in a research space with varied solution approaches, potentially signaling higher innovation potential.

### 3.3.3 NOVELTY

Novelty assesses the originality and innovation level of generated ideas by analyzing their similarity to existing literature while considering publication timing. Each generated idea first undergoes distillation into a searchable representation $\mathcal{Q}$; we then compute the cosine similarity $\sigma_i$ between $\mathcal{Q}$ and each matched paper $P_i$ in the set $\{P_i\}_{i=1}^n$ using the embedding function $\mathcal{E}(\cdot)$.

**Step 1: Time-Recency Factor** We incorporate temporal considerations through publication years. Let $y_i$ denote the publication year of paper $P_i$, $Y_{\max}$ the most recent publication year among reference papers, and $Y_{\text{now}}$ the current year. The time-recency factor is

$$t_i = \frac{\max(0,\ y_i - Y_{\max})}{Y_{\text{now}} - Y_{\max}} \in [0,1], \tag{6}$$

so that $t_i = 0$ for papers predating the reference set and $t_i \to 1$ for newly published work.

**Step 2: Paper Selection and Weighting** To focus on the most pertinent prior work we keep only the top-$k$ most similar papers; $k$ therefore fixes the evaluation scope and can be tuned to trade recall for precision. We then assign similarity-based weights

$$w_i = \frac{\sigma_i^{\beta}}{\sum_{j=1}^n \sigma_j^{\beta}}, \qquad \sum_{i=1}^n w_i = 1, \tag{7}$$

where the sharpness parameter $\beta > 0$ controls how strongly the most similar papers dominate: larger $\beta$ concentrates weight on the highest-similarity matches, whereas $\beta = 1$ yields a softmax-like smoothing.

**Step 3: Novelty Score Computation** The overall novelty score penalizes strong resemblance to older work while rewarding alignment with recent literature:

$$\text{Penalty} = \lambda \sum_{i=1}^n w_i\,\sigma_i^2\,(1-t_i)^2, \quad \text{Bonus} = \alpha \sum_{i=1}^n w_i\,\sigma_i\,t_i^2, \tag{8}$$

$$\text{N}(\mathcal{Q}) = 1 - \text{Penalty} + \text{Bonus}, \tag{9}$$

where $\lambda$ weights the similarity-to-old-work penalty and $\alpha$ weights the recency bonus. Setting $\lambda > \alpha$ makes the metric conservative—discouraging overlap even with very recent papers—whereas $\alpha > \lambda$ encourages building on the latest advances. The resulting score lies in $[0,1]$, with higher values indicating greater novelty.

## 4 DATASET CONSTRUCTION

We built our dataset by systematically collecting computer science papers from arXiv (Cornell University, 1991) across multiple years (details in Appendix A.5). As shown in Table 1, we used large language models to automatically extract and structure content across six key dimensions: tasks, gaps, contributions, methods, experiments, and literature reviews. The dataset contains 3,353 papers covering eight research tasks across four major CS fields—Machine Learning (Reinforcement Learning, Representation Learning), NLP (Classification, Machine Translation), Computer Vision (Object Detection, Semantic Segmentation), and Distributed Computing (Consensus Algorithm, Data Processing). Each paper includes complete metadata, citation metrics (ranging from 0 to 21,752 total citations), and structured research content. We capture tasks as problem statements, gaps as limitation lists, contributions as solutions addressing those gaps, and methods as input-method-output-detail quadruplets. Full extraction procedures are in Appendix A.8.

## 5 EXPERIMENTAL SETUP

All experiments use GPT-4o-mini (OpenAI, 2024) for text generation and text-embedding-3-small (OpenAI, 2024) for embeddings. We use the OpenAlex academic database (Priem et al., 2022) to retrieve papers and match against generated ideas through semantic search.

| Field | Task | Papers | Citations Range | Cit/Year Range |
|---|---|---|---|---|
| LG | Reinforcement Learning | 618 | 0–184 | 0.0–51.98 |
| | Representation Learning | 530 | 0–1070 | 0.0–299.02 |
| CL/NLP | Classification | 594 | 2–623 | 0.54–244.42 |
| | Machine Translation | 325 | 0–1107 | 0.0–309.6 |
| CV | Object Detection | 348 | 4–21752 | 1.53–5086.38 |
| | Semantic Segmentation | 323 | 0–729 | 0.0–275.92 |
| DC | Consensus Algorithm | 119 | 0–113 | 0.0–28.64 |
| | Data Processing | 496 | 0–413 | 0.0–133.49 |
| | **Total** | **3353** | **0–21752** | **0.0–5086.38** |

Table 1: Dataset statistics across eight research tasks with paper counts and citation metrics. Fields: LG (Machine Learning), CL/NLP (Computational Linguistics/Natural Language Processing), CV (Computer Vision), DC (Distributed Computing).

## 5.1 ABLATION STUDIES

We conduct three systematic experiments to evaluate our approach's components. **Baseline** performs idea generation without enhanced representations or paper embeddings, using CoI paper analysis agent and idea generation agent (Li et al., 2024). **Enhanced** incorporates paper representations but uses CoI idea generation agent. **Full System** implements the complete system with both paper representation and full representation-based idea generation. For each topic, we generated 780 ideas by combining papers with the top 40 citations, with 2 papers per combination. Then, we evaluate each method using 3 metrics from Section 3.3. Proportion (**Prop.**) measures the percentage of ideas in each similarity range: high ($\geq 0.7$), mid ($0.3 - 0.7$), and low ($\leq 0.3$). Unique Paper Ratio (**U.R.**) quantifies the percentage of unique papers matched within each similarity category. **Novelty** assesses idea originality using weighted similarity scores that penalize matches to older papers while rewarding alignment with recent publications. We analyze correlations among log-transformed citation counts $\log(C + 1)$, similarity scores $S$, and novelty scores $N$, using logarithmic transformation to handle skewed citation distributions and reduce outlier influence.

## 5.2 STUDY ON IDEA COMPOSITION

To analyze idea composition, we first classify papers into three categories based on their primary contribution: experimental papers (novel methods/algorithms), resource papers (datasets/benchmarks/tools), and positional papers (surveys/theoretical analyses/position statements). Experimental papers drive methodological innovation through empirical validation, resource papers facilitate reproducible research through standardized frameworks, and positional papers establish conceptual foundations but offer limited technical innovations. Since positional papers provide minimal practical insights for cross-pollination, we focus on combining resource and experimental papers with different ratios: 40:0 (experimental only), 5:1, 3:1, and 1:1 across the 8 research tasks. For pairing, resource papers serve as base studies while experimental papers act as innovation sources. When pairs contain only experimental studies, we optimize base-source role assignment for cross-pollination effectiveness.

# 6 RESULTS ANALYSIS

## 6.1 ABLATION STUDIES

Table 2 reveals how our idea generation approaches progressively improve. For high similarity ideas (those scoring $\geq 0.7$), we observe a clear upward trend, with a **41% relative gain** overall. What's particularly interesting is that the Full System achieves this better alignment with established research directions while keeping novelty scores steady around 0.93, effectively balancing innovation with relevance. The Full System also delivers notable improvements in idea diversity. For high similarity ideas, unique paper matching reaches **78.4%**, outperforming both the Baseline (76.9%) and Enhanced approach (70.6%). Even more striking is what happens with low similarity ideas: their unique ratios surge from 35.8% in the Baseline to **57.3%** in the Full System—a remarkable **60% improvement**. These gains hold up consistently across different domains, with particularly strong performance in areas such as the "Consensus Algorithm" and the "Machine Translation." This con-

| Methods | Topic | High Similarity (≥ 0.7) | | | Mid Similarity (0.3 − 0.7) | | | Low Similarity (≤ 0.3) | | |
|---|---|---|---|---|---|---|---|---|---|---|
| | | Prop. (%) | U.R. (%) | Novelty | Prop. (%) | U.R. (%) | Novelty | Prop. (%) | U.R. (%) | Novelty |
| Baseline | Classification | 17.4 | 55.9 | 0.344 | 82.6 | 62.6 | 0.517 | 0.0 | 0.0 | N/A |
| | Consensus Algorithm | 7.8 | 68.9 | 0.344 | 89.6 | 50.8 | 0.537 | 2.6 | 25.0 | 0.989 |
| | Data Processing | 13.3 | 79.8 | 0.351 | 84.4 | 74.8 | 0.532 | 2.3 | 44.4 | 0.978 |
| | Machine Translation | 2.2 | 76.5 | 0.326 | 95.5 | 75.2 | 0.575 | 2.3 | 22.2 | 0.989 |
| | Object Detection | 8.6 | 80.6 | 0.377 | 91.0 | 73.1 | 0.536 | 0.4 | 100.0 | 0.931 |
| | Reinforcement Learning | 7.1 | 85.5 | 0.330 | 92.4 | 79.6 | 0.538 | 0.5 | 75.0 | 0.963 |
| | Representation Learning | 2.9 | 82.6 | 0.358 | 95.7 | 84.9 | 0.542 | 1.3 | 20.0 | 0.990 |
| | Semantic Segmentation | 6.9 | 85.2 | 0.361 | 93.1 | 77.8 | 0.530 | 0.0 | 0.0 | N/A |
| | **Mean** | **8.3** | **76.9** | **0.349** | **90.5** | **72.4** | **0.538** | **1.2** | **35.8** | **0.973** |
| Enhanced | Classification | 19.7 | 55.8 | 0.348 | 80.1 | 68.8 | 0.519 | 0.1 | 100.0 | 0.897 |
| | Consensus Algorithm | 7.8 | 67.2 | 0.349 | 89.7 | 60.4 | 0.538 | 2.4 | 26.3 | 0.981 |
| | Data Processing | 14.5 | 81.4 | 0.361 | 83.6 | 77.6 | 0.532 | 1.9 | 46.7 | 0.965 |
| | Machine Translation | 2.4 | 73.7 | 0.346 | 95.5 | 70.1 | 0.575 | 2.1 | 25.0 | 0.989 |
| | Object Detection | 7.7 | 81.7 | 0.380 | 91.9 | 70.0 | 0.540 | 0.4 | 33.3 | 1.000 |
| | Reinforcement Learning | 8.2 | 60.9 | 0.339 | 90.5 | 77.8 | 0.536 | 1.3 | 50.0 | 0.969 |
| | Representation Learning | 4.6 | 69.4 | 0.375 | 94.6 | 81.4 | 0.545 | 0.8 | 66.7 | 0.972 |
| | Semantic Segmentation | 8.1 | 74.6 | 0.363 | 91.9 | 69.7 | 0.527 | 0.0 | 0.0 | N/A |
| | **Mean** | **9.1** | **70.6** | **0.358** | **89.7** | **72.0** | **0.539** | **1.1** | **43.5** | **0.968** |
| Full System | Classification | 20.9 | 57.7 | 0.339 | 79.1 | 67.9 | 0.511 | 0.0 | 0.0 | N/A |
| | Consensus Algorithm | 10.1 | 81.0 | 0.350 | 87.9 | 68.8 | 0.537 | 1.9 | 33.3 | 0.975 |
| | Data Processing | 18.2 | 79.6 | 0.352 | 80.9 | 82.3 | 0.528 | 0.9 | 42.9 | 0.987 |
| | Machine Translation | 5.4 | 85.7 | 0.344 | 93.6 | 70.6 | 0.560 | 1.0 | 50.0 | 0.985 |
| | Object Detection | 9.6 | 69.3 | 0.361 | 90.1 | 75.9 | 0.540 | 0.3 | 100.0 | 0.946 |
| | Reinforcement Learning | 8.1 | 84.1 | 0.333 | 91.8 | 80.4 | 0.533 | 0.1 | 100.0 | 1.000 |
| | Representation Learning | 4.7 | 91.9 | 0.350 | 94.2 | 86.1 | 0.540 | 1.0 | 25.0 | 0.993 |
| | Semantic Segmentation | 10.5 | 85.4 | 0.358 | 89.4 | 76.0 | 0.522 | 0.1 | 100.0 | 0.876 |
| | **Mean** | **11.7** | **78.4** | **0.352** | **87.6** | **76.8** | **0.534** | **0.6** | **57.3** | **0.966** |

Table 2: **Comparison of Three Methods: Similarity Distribution and Novelty Scores.** 780 ideas per topic across 8 topics. Methods: Baseline (without enhanced representations), Enhanced (with paper embeddings), Full System (with paper representation and idea generation). Prop. = proportion in similarity range; U.R. = unique paper ratio; Novelty = mean novelty score.

sistency suggests that the full system has learned generalizable and feasible patterns for reliable idea generation rather than simply picking up domain-specific tricks.

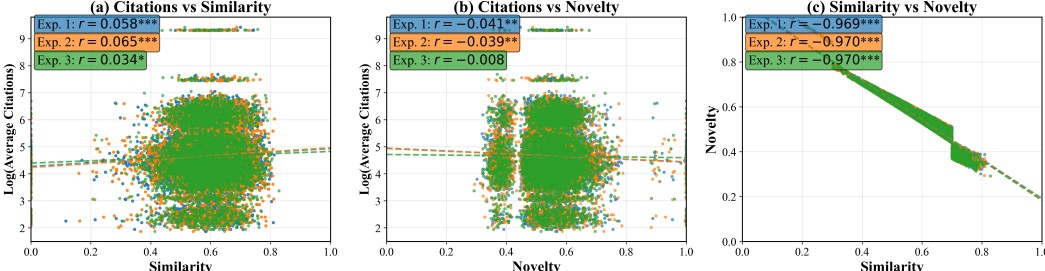

Figure 4: **Correlation analysis of citations, similarity, and novelty across three conditions.** (a) Correlationship between log-transformed average citations and similarity scores. (b) Correlationship between log-transformed average citations and novelty scores. (c) Correlationship between similarity and novelty scores. Exp. 1: CoI Generation; Exp. 2: Representation-Enhanced CoI; Exp. 3: Full System. Linear fits with correlation coefficients (* $p < 0.05$, ** $p < 0.01$, *** $p < 0.001$). Strong negative correlation in (c) indicates inverse similarity-novelty relationship.

## 6.2 CORRELATION ANALYSIS

Correlation analysis reveals distinct patterns among log average citation, similarity, and novelty across the three experimental conditions. Citation and similarity show consistently weak but positive correlations across all methods. Later experiments confirm this positive relationship between citation and similarity. Citation and novelty show negligible correlations, indicating that novel ideas with high similarity scores don't necessarily come from highly-cited papers. In contrast, similarity and novelty exhibit extremely strong negative correlations across all methods, confirming the expected inverse relationship where higher similarity corresponds to lower novelty.

| Topic | 40 : 0 (%) | 5 : 1 (%) | 3 : 1 (%) | 1 : 1 (%) |
|---|---|---|---|---|
| Classification | 3.21 | 4.12 | 5.00 | 4.25 |
| Consensus Algorithm | 4.38 | 6.88 | 5.93 | 9.34 |
| Data Processing | 5.38 | 4.75 | 4.50 | 8.20 |
| Machine Translation | 1.80 | 1.67 | 1.63 | 1.97 |
| Object Detection | 4.10 | 2.95 | 4.62 | 4.10 |
| Reinforcement Learning | 3.38 | 2.88 | 3.54 | 3.55 |
| Representation Learning | 3.17 | 3.12 | 2.76 | 1.58 |
| Semantic Segmentation | 3.36 | 3.80 | 2.85 | 1.99 |
| **Mean** | **3.60** | **3.77** | **3.85** | **4.37** |

Table 3: High-similarity (score $\geq$ 0.7) proportions for each topic under four experimental-resource paper ratios in generating idea.

### 6.3 IDEA COMPOSITION

Table 3 shows high-similarity idea proportions increase across experimental-resource paper ratios (40:0, 5:1, 3:1, 1:1). While idea quality drops when using less impactful papers compared to ablation studies with top 40 papers, this is less critical for composition studies which examine research focus shifts across tasks. Topic responses vary significantly: Consensus Algorithm and Data Processing benefit from resource papers (Consensus Algorithm doubles from 4.38% to 9.34%), while Representation Learning and Semantic Segmentation decline with more resource papers. Machine Translation remains consistently low, suggesting resistance to cross-domain pollination. These results indicate the need for domain-specific optimization of paper ratios to maximize cross-pollination effectiveness.

## 7 RELATED WORKS

Recent advances in AI-driven scientific discovery span the complete research pipeline. For hypothesis generation, systems like SGA (Ma et al., 2024), Chain-of-Ideas (Li et al., 2024), and SciAgents (Ghafarollahi & Buehler, 2024b) employ LLM frameworks and simulated researcher teams. Experimental design leverages search algorithms, with ChemReasoner using hierarchical trees (Sprueill et al., 2024) and MC-NEST applying Monte Carlo Tree Search (Rabby & Lee, 2025). Automated experimentation systems include ProtAgents for protein design (Ghafarollahi & Buehler, 2024a), Sparks for materials discovery (Ghafarollahi & Buehler, 2025), and SARA's active-learning loops for synthesis (Ament et al., 2021). Analysis frameworks integrate formal validation through theorem provers (Quan et al., 2024) and knowledge graphs (Ghafarollahi & Buehler, 2024b). Complete workflow automation is demonstrated by AI Scientist (Lu et al., 2024) and AI Scientist-v2 (Yamada et al., 2025), achieving competitive results in machine learning, while industry initiatives like Google DeepMind's AI Co-Scientist accelerate biomedical discovery. ScienceAgentBench provides rigorous evaluation benchmarks for assessing these agents' capabilities across scientific tasks (Chen et al., 2025).

## 8 CONCLUSION

This paper introduces a novel framework for automated research idea generation through literature-driven cross-pollination. Our approach generates more relevant and implementable ideas by using structured paper representations and systematically bridging gaps between existing studies. Experiments across 3,353 papers from eight computer science domains demonstrate consistent improvements: our system achieves a 41% relative increase in high-similarity ideas while maintaining stable novelty scores. The strong negative correlation between similarity and novelty confirms successful navigation of the relevance-innovation trade-off. Analysis reveals that cross-pollination effectiveness varies by experimental-resource paper ratios, with domains like Consensus Algorithm showing nearly doubled high-similarity performance with high resource proportions, while others remain resistant to this approach. Unlike current methods, our framework provides explicit traceability and grounding in existing literature, enabling systematic identification of research gaps. Future work could explore domain-specific optimization and develop more sophisticated metrics for evaluating practical implementability.

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

# A    APPENDIX

## A.1    SENSITIVITY ANALYSIS

To evaluate the robustness of our proposed novelty computation, we conducted comprehensive sensitivity analyses across different parameter configurations. We tested 150 parameter combinations on 432 research ideas, varying $\beta$ (1-5), $\lambda$ (1-5), and $\alpha$ (0-5). Based on 1000 bootstrap iterations and permutation tests, we report the following results.

### A.1.1    PARAMETER EFFECTS

Table 4 summarizes the individual effects of each parameter on the novelty scores.

| Parameter | Result | 95% CI | P-Value | Interpretation |
|---|---|---|---|---|
| $\lambda$ (penalty) effect | 0.50 range | [0.45, 0.55] | $p < 0.001$ | Highly significant driver |
| $\alpha$ (bonus) effect | 0.50 range | [0.44, 0.56] | $p < 0.001$ | Highly significant driver |
| $\beta$ (weight) effect | 0.13 range | [0.11, 0.15] | $p < 0.01$ | Minor but significant |

Table 4: Parameter effects on novelty scores.

Among the three parameters, only $\lambda$ and $\alpha$ have significant effects on novelty scores. Together, these two parameters account for essentially all the variation in the metric, with $\beta$ contributing only 13%. The interplay between $\lambda$ and $\alpha$ shapes the metric's behavior fundamentally. When $\lambda$ is high and $\alpha$ is low, the metric becomes overly conservative, which generates scores around 0.35. Conversely, when $\lambda$ is low and $\alpha$ is high, the metric becomes too permissive, with most scores approaching 1.00. For balanced and interpretable results, we recommend keeping $\lambda$ and $\alpha$ within 1-2 units of each other.

### A.1.2    CONFIGURATION COMPARISONS

Table 5 presents comparisons of different parameter configurations against the baseline configuration ($\beta = 3$, $\lambda = 3$, $\alpha = 3$).

| Configuration | Result | 95% CI | P-Value | Interpretation |
|---|---|---|---|---|
| $\beta = 3, \lambda = 4, \alpha = 1$ | $-0.28$ | $[-0.30, -0.26]$ | $p < 0.001$ | Significantly stricter |
| $\beta = 2, \lambda = 2, \alpha = 2$ | $-0.10$ | $[-0.12, -0.08]$ | $p < 0.001$ | Moderately stricter |
| $\beta = 2, \lambda = 1, \alpha = 4$ | $+0.05$ | $[+0.03, +0.07]$ | $p < 0.01$ | Slightly more lenient |

Table 5: Configuration comparisons (vs. $\beta = 3$, $\lambda = 3$, $\alpha = 3$).

### A.1.3    RANK STABILITY

To assess whether the ranking of ideas remains stable across different parameter settings, we computed the Spearman correlation coefficient.

We found strong statistical robustness; these results would replicate across different idea samples. While different parameter combinations produce moderately stable rankings overall ($\rho = 0.787$), the exact ordering does shift. In practical terms, this means the top-ranked ideas consistently remain near the top, but their precise positions within the top 20 will vary. The majority of ideas (71%) show score variations exceeding 0.5 across different parameter settings. This instability is particularly

| Test | Result | P-Value | Interpretation |
|------|--------|---------|----------------|
| Spearman correlation | 0.787 | $p < 0.001$ | Rankings are stable, not random |

Table 6: Rank stability analysis.

pronounced for ideas scoring in the middle range (0.4-0.7), where rankings can shift dramatically. By contrast, ideas with extreme scores (either clearly above 0.8 or below 0.3) demonstrate greater stability across parameter choices.

## A.2 HUMAN ANNOTATION RESULTS

### A.2.1 RECRUITMENT CRITERIA AND VERIFICATION

We designed our recruitment process to ensure that each of the 8 AI/ML topics was evaluated by genuine experts in that specific area. For each topic, we recruited two independent reviewers in Computer Science, Artificial Intelligence, Machine Learning, or closely related fields respectively, and who had actively published in the topic area they were assigned to review.

Beyond basic credentials, we looked for reviewers with recent engagement in their fields—specifically, peer-reviewed publications within the past three years and current or recent affiliations with recognized research institutions or industry research labs. This ensured our reviewers were familiar with current state-of-the-art methods and could provide informed assessments.

### A.2.2 VERIFICATION PROCESS

We used a three-stage approach to verify reviewer qualifications and build trust in our evaluation process:

1. **Database Search**: We identified potential reviewers by systematically searching academic databases including Google Scholar, DBLP, Semantic Scholar, and arXiv. We focused on researchers who had published as first or corresponding authors in top-tier conferences and journals relevant to each topic. To complement this database search, we also gathered peer recommendations from established researchers in each domain.

2. **Credential Verification**: We independently confirmed each candidate's publication record by checking official conference and journal websites, reading their papers to verify substantive contributions to the topic area, and ensuring they had at least two peer-reviewed publications directly relevant to their assigned topic. We also verified institutional affiliations through university websites, ORCID profiles, and professional networks to confirm current research activity.

3. **Screening Interviews**: We conducted brief screening interviews (15-20 minutes) with each candidate to assess their depth of knowledge in the specific topic, their familiarity with current methods, and their understanding of what the evaluation task would involve. Candidates also provided their CVs and publication lists for our review.

### A.2.3 TOPIC ASSIGNMENT AND INDEPENDENCE

Once qualified, we assigned reviewers to topics matching their publication expertise, with two independent experts per topic. To ensure unbiased assessments, reviewers didn't know who the other reviewer for their topic was and were instructed not to discuss their evaluations with anyone.

### A.2.4 REVIEWER POOL CHARACTERISTICS

Table 7 summarizes the characteristics of our final reviewer pool.

### A.2.5 TRAINING AND QUALITY CONTROL

Before beginning the actual evaluation, all reviewers participated in an hour training session where we walked through the evaluation criteria, rating scales, and annotation guidelines in detail. Fol-

| Metric | Value |
|---|---|
| Total Reviewers | 16 PhD-level experts |
| Reviewers per Topic | 2 independent annotators |
| Publication Range | 2-15+ peer-reviewed papers |
| Geographic Distribution | 3 countries (North America, Asia) |
| Institutional Diversity | 4 different universities/research institutions |

Table 7: Reviewer pool characteristics.

lowing training, each reviewer completed a calibration phase, annotating five practice items and receiving feedback to ensure they understood the task correctly. Throughout the annotation process, reviewers had access to a dedicated point of contact for any questions or clarifications.

We maintained detailed documentation throughout, including records of all candidate contacts, verification materials, training completion, and timestamped annotation submissions. This documentation provides an audit trail and enables potential replication of our study.

### A.2.6 ANNOTATION RESULTS

Table 8 presents the inter-rater reliability metrics across all eight topics.

| Topic | N | M1 (SD1) | M2 (SD2) | % Agree | $\kappa$ | $r / \rho$ | ICC |
|---|---|---|---|---|---|---|---|
| Consensus Algorithm | 78 | 3.99 (0.11) | 4.00 (0.00) | 98.7 | 0.000 | — | — |
| Data Processing | 78 | 4.00 (0.00) | 4.00 (0.00) | 100.0 | nan | — | — |
| Machine Translation | 78 | 3.96 (0.19) | 3.97 (0.16) | 96.2 | 0.381 | 0.389*** | 0.386 |
| Object Detection | 78 | 3.95 (0.22) | 3.99 (0.11) | 93.6 | $-0.021$ | $-0.026$ | $-0.027$ |
| Reinforcement Learning | 78 | 4.00 (0.00) | 4.00 (0.00) | 100.0 | nan | — | — |
| Representation Learning | 78 | 4.00 (0.00) | 4.00 (0.00) | 100.0 | nan | — | — |
| Semantic Segmentation | 78 | 3.97 (0.16) | 3.99 (0.11) | 98.7 | 0.661 | 0.703*** | 0.664 |
| Sentiment Analysis | 78 | 3.86 (0.35) | 3.99 (0.20) | 87.2 | 0.248 | 0.350** | 0.241 |
| Text Classification | 78 | 3.95 (0.22) | 3.97 (0.16) | 94.9 | 0.310 | 0.330** | 0.312 |
| **Overall** | 702 | 3.96 (0.19) | 3.99 (0.11) | 96.6 | 0.281 | 0.324*** | 0.279 |

Table 8: Human annotation results by topic. M1 = Annotator 1 Mean, M2 = Annotator 2 Mean, $\kappa$ = Cohen's Kappa, $r$ = Pearson correlation, $\rho$ = Spearman correlation, ICC = Intraclass Correlation Coefficient ICC(2,1). No variance gives nan. *** $p < 0.001$, ** $p < 0.01$.

The success is reflected in the strong inter-rater reliability we observed. Overall, the two independent reviewers for each topic agreed 96.6% of the time, with three topics (Data Processing, Reinforcement Learning, and Representation Learning) showing perfect agreement. The statistically significant reliability coefficients across most topics confirm that we successfully identified qualified experts capable of providing consistent evaluations across all 702 assessment items.

### A.2.7 NOVELTY RATING SCALE

Annotators should pretend they only know studies published before 2022, as the parent papers used in idea generation are from 2021 and 2020. Novelty refers to the degree to which a research idea introduces original concepts, methods, or insights that meaningfully advance beyond existing work. Novelty encompasses both the uniqueness of the approach and its potential to impact the field, ranging from incremental improvements to paradigm-shifting innovations.

- **5: Paradigm-shifting innovation** — Opening new research directions with fundamentally new concepts challenging existing paradigms. No significant prior work on this specific problem.
- **4: Significant innovation** — Clear advancement through novel combination of concepts in non-obvious ways. Addresses known gaps innovatively with clear differentiation from existing methods.
- **3: Solid incremental contribution** — Meaningful extension of existing approaches. Combines known techniques sensibly and addresses specific limitations of prior work.

- **2: Minor variation** — Variation of existing approaches or straightforward application to new domain. Limited differentiation with predictable extension, primarily representing an engineering contribution.

- **1: Largely derivative** — Trivial differences or direct replication with minor parameter changes. No clear advancement and questionable research contribution.

- **0: Exact replication** — Common knowledge that has been thoroughly explored. No distinguishable contribution and not publishable.

Importantly, our annotators were not provided with predefined answers or reference materials during the evaluation process. Instead, they relied solely on their own domain expertise and understanding of their respective fields to make judgments. As with other expert-driven evaluations, these annotations are inherently subjective and should be interpreted as informed assessments rather than definitive ground truth. **For this reason, we recommend placing greater confidence in our proposed objective novelty evaluation, which provides measurable and reproducible metrics, over subjective assessments.**

## A.3 MODEL COMPARISON EXPERIMENTS

To address concerns about temporal confounding and model dependency, we conducted experiments across three different GPT models with different knowledge cutoffs: GPT-3.5-turbo (training data cutoff: January 2022), GPT-4.1 (training data cutoff: June 2024), and GPT-4o-mini (our primary model).

### A.3.1 GPT-3.5-TURBO RESULTS

Table 9 presents the full results using GPT-3.5-turbo as the backbone model.

| Topic | High_Prop | High_U.R. | High_Novel | Mid_Prop | Mid_U.R. | Mid_Novel | Low_Prop | Low_U.R. | Low_Novel |
|---|---|---|---|---|---|---|---|---|---|
| Consensus Algorithm | 20.51 | 56.25 | 0.35 | 78.21 | 68.9 | 0.54 | 1.28 | 100.00 | 0.98 |
| Data Processing | 19.23 | 80.00 | 0.35 | 75.64 | 83.1 | 0.53 | 5.13 | 75.0 | 0.99 |
| Machine Translation | 1.28 | 0.00 | 0.34 | 98.72 | 71.4 | 0.56 | 0.00 | 0.00 | NaN |
| Object Detection | 14.10 | 81.82 | 0.36 | 85.90 | 76.1 | 0.54 | 0.00 | 0.00 | NaN |
| Reinforcement Learning | 10.26 | 75.00 | 0.33 | 88.46 | 79.7 | 0.53 | 1.28 | 100.00 | 1.00 |
| Representation Learning | 2.56 | 100.00 | 0.35 | 96.15 | 85.3 | 0.54 | 1.28 | 100.00 | 0.99 |
| Semantic Segmentation | 15.38 | 100.00 | 0.36 | 84.62 | 75.8 | 0.52 | 0.00 | 0.00 | NaN |
| Sentiment Analysis | 16.67 | 84.62 | 0.34 | 83.33 | 67.7 | 0.51 | 0.00 | 0.00 | NaN |
| Text Classification | 2.56 | 100.00 | 0.34 | 96.15 | 67.6 | 0.51 | 1.28 | 100.00 | 0.98 |
| **Mean** | **11.40** | **75.30** | **0.35** | **87.46** | **75.1** | **0.53** | **1.14** | **63.89** | **0.99** |

Table 9: Full system results with GPT-3.5-turbo.

### A.3.2 GPT-4.1 RESULTS

Table 10 presents the full results using GPT-4.1 as the backbone model.

| Topic | High_Prop | High_U.R. | High_Novel | Mid_Prop | Mid_U.R. | Mid_Novel | Low_Prop | Low_U.R. | Low_Novel |
|---|---|---|---|---|---|---|---|---|---|
| Consensus Algorithm | 21.79 | 65.00 | 0.33 | 76.92 | 70.0 | 0.56 | 1.28 | 100.00 | 0.98 |
| Data Processing | 23.08 | 57.14 | 0.34 | 74.36 | 81.0 | 0.55 | 2.56 | 50.0 | 0.99 |
| Machine Translation | 3.85 | 100.00 | 0.32 | 93.59 | 68.8 | 0.58 | 2.56 | 100.00 | 0.99 |
| Object Detection | 25.64 | 95.65 | 0.35 | 74.36 | 77.6 | 0.56 | 0.00 | 0.00 | NaN |
| Reinforcement Learning | 8.97 | 50.00 | 0.31 | 91.03 | 82.5 | 0.55 | 0.00 | 0.00 | NaN |
| Representation Learning | 2.56 | 100.00 | 0.33 | 96.15 | 86.7 | 0.56 | 1.28 | 100.00 | 0.99 |
| Semantic Segmentation | 10.26 | 100.00 | 0.34 | 89.74 | 74.3 | 0.54 | 0.00 | 0.00 | NaN |
| Sentiment Analysis | 12.82 | 100.00 | 0.32 | 84.62 | 69.7 | 0.53 | 2.56 | 100.00 | 0.88 |
| Text Classification | 0.00 | 0.00 | NaN | 98.72 | 66.2 | 0.53 | 1.28 | 100.00 | 0.95 |
| **Mean** | **12.11** | **74.20** | **0.33** | **86.61** | **75.2** | **0.55** | **1.28** | **72.22** | **0.96** |

Table 10: Full system results with GPT-4.1.

### A.3.3 BASELINE COMPARISON ACROSS MODELS

Table 11 compares the baseline (COI), enhanced, and full system methods across all three models.

| Model | Method | High_Prop | High_U.R. | Mid_Prop | Mid_U.R. | Low_Prop | Low_U.R. |
|---|---|---|---|---|---|---|---|
| | COI | 8.26 | 69.25 | 90.31 | 67.1 | 1.42 | 47.22 |
| GPT-3.5 | Enhanced | 9.69 | 72.74 | 89.03 | 71.0 | 1.28 | 51.39 |
| | Full System | **11.40** | **75.30** | **87.46** | **75.1** | **1.14** | **63.89** |
| | COI | 8.83 | 67.04 | 89.74 | 67.4 | 1.42 | 55.56 |
| GPT-4.1 | Enhanced | 10.25 | 70.70 | 88.46 | 71.3 | 1.28 | 65.28 |
| | Full System | **12.11** | **74.20** | **86.61** | **75.2** | **1.28** | **72.22** |
| | COI | 8.3 | 76.9 | 90.5 | 72.4 | 1.2 | 35.8 |
| GPT-4o | Enhanced | 9.1 | 70.6 | 89.7 | 72.0 | 1.1 | 43.5 |
| | Full System | **11.7** | **78.4** | **87.6** | **76.8** | **0.6** | **57.3** |

Table 11: Baseline comparison across different GPT models.

### A.3.4 STATISTICAL SIGNIFICANCE OF OVERALL ADVANTAGE

Table 12 shows the relative improvement achieved by our Full System over the baseline across all three models.

| Model | Baseline → Full System | Relative Improvement |
|---|---|---|
| GPT-3.5 | 8.26 → 11.40 | +38.0% |
| GPT-4.1 | 8.83 → 12.11 | +37.1% |
| GPT-4o | 8.3 → 11.7 | +41.0% |

Table 12: Relative improvement in High Similarity Proportion.

The overall advantage of our Full System is statistically significant as demonstrated in the mean results. On GPT-4o, the High Similarity Proportion improves from **8.3% (Baseline) to 11.7% (Ours)**, representing a **41% relative improvement**. This consistent improvement is observed across all three LLM backbones.

### A.3.5 DISCUSSION

The similar performance across GPT-3.5-turbo, GPT-4.1, and GPT-4o-mini (High Similarity Proportion: 11.40%, 12.11%, and 11.7% respectively) is *not* surprising given our framework design. In our framework, **models are NOT relying on their training data to generate ideas**. Instead, we provide explicit paper information as input, and all models are prompted to synthesize and reassemble this information rather than generate ideas from their parametric knowledge. This design choice is precisely why knowledge cutoff effects are largely irrelevant to our methodology—our method's effectiveness stems from its ability to better guide the model in assembling novel ideas, rather than merely exploiting the model's existing knowledge of future publications.

The consistency across models with vastly different training cutoffs (January 2022 for GPT-3.5 vs. June 2024 for GPT-4.1) provides strong evidence that our comparative results are not confounded by the model's prior knowledge of papers published after the parent papers used in idea generation.

### A.4 HUMAN EVALUATION ON AUTO-GENERATED PAPER IDEA REPRESENTATION

We conducted a comprehensive evaluation comparing auto-generated JSON paper representations against human-revised versions across 19 computer science research papers. To quantify the similarity between automated and human-validated representations, we computed cosine similarity scores on vectorized representations of the extracted features, defined as:

$$\cos(\theta) = \frac{\mathbf{A} \cdot \mathbf{B}}{||\mathbf{A}|| \cdot ||\mathbf{B}||} = \frac{\sum_{i=1}^{n} A_i B_i}{\sqrt{\sum_{i=1}^{n} A_i^2} \cdot \sqrt{\sum_{i=1}^{n} B_i^2}} \tag{10}$$

where $\mathbf{A}$ and $\mathbf{B}$ represent the feature vectors of the automated and human-validated representations respectively, providing a normalized measure of alignment ranging from 0 (completely dissimilar) to 1 (identical). The analysis reveals both the strengths and limitations of current automated extraction methodologies.

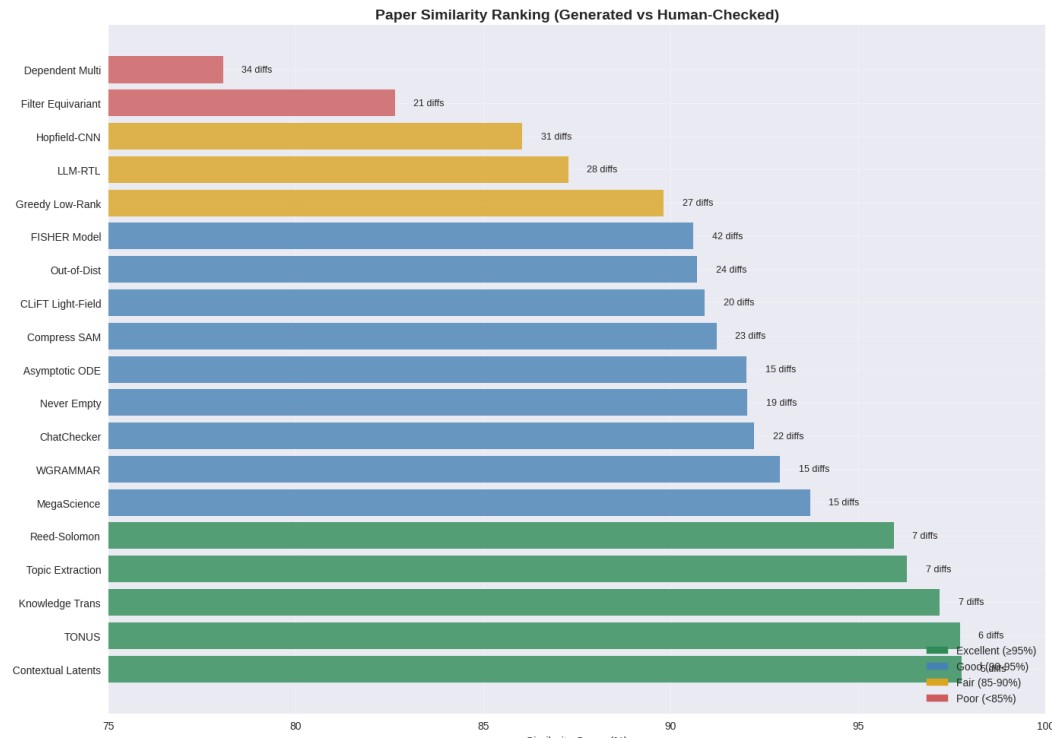

Figure 5: Paper similarity ranking comparing automated generation against human-validated representations. Papers are color-coded by performance level: green (excellent, $\geq$95%), blue (good, 90-95%), gold (fair, 85-90%), and red (poor, <85%). Numbers indicate total differences found.

### A.4.1 OVERALL PERFORMANCE METRICS

The automated system achieved an average similarity score of 91.32% when compared to human-validated representations, with no papers achieving perfect matches (100% similarity). Figure 5 illustrates the complete ranking of papers by similarity score, revealing a right-skewed distribution with 5 papers (26.3%) exceeding 95% similarity, 11 papers (57.9%) falling between 85-95%, and 3 papers (15.8%) below 85%.

The highest-performing papers demonstrated similarity scores of 97.77% (*From One to More Contextual Part Latents*), 97.71% (*TONUS Neuromorphic human pose estimation*), and 97.17% (*Uncertainty-Aware Knowledge Transformers*). Conversely, the most challenging papers for automated extraction were *Dependent Multiplicities in Dependent Linear Type Theory* (78.06%) and *Filter Equivariant Functions* (82.64%). Figure 6 provides a detailed breakdown of extraction performance across different paper sections. The heatmap visualization reveals systematic patterns in automated extraction capabilities, with clear performance variations between structural elements of research papers.

### A.5 DATA RETRIEVAL

Papers were retrieved from the ArXiv preprint repository using the Python ArXiv API client. Search queries combined domain-specific terms with temporal constraints, formatted as $q \wedge \text{submittedDate} \in [y0101, y1231]$ for query $q$ and year $y$. The system retrieved up to $\min(3n, 2000)$ results per query, where $n$ represents the target paper count. A two-tier search strategy provided robustness: primary searches used date-filtered queries, while a fallback mechanism employed simple queries with post-retrieval year filtering when parsing errors occurred.

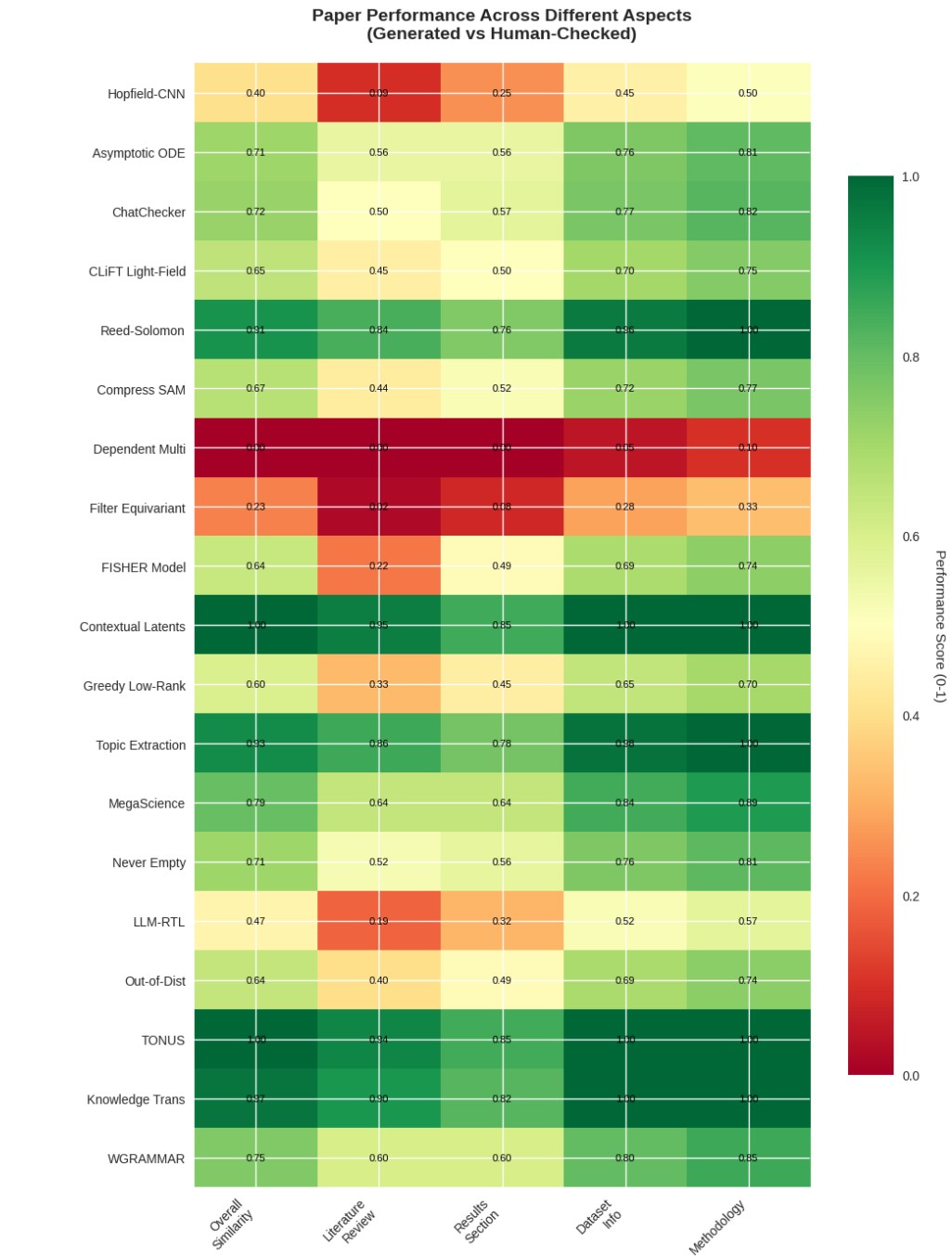

Figure 6: Performance heatmap showing automated extraction accuracy across different paper sections. Darker green indicates higher performance, while red indicates areas requiring improvement. Each cell shows the normalized performance score (0-1 scale).

### A.5.1 PROCESSING PIPELINE

The acquisition pipeline utilized several key tools for document processing. The `requests` library handled PDF downloads with 30-second timeouts. PyPDF2 extracted text from PDF documents page-by-page. The system maintained a persistent cache to prevent redundant downloads and text extractions across multiple executions.

Papers underwent validation to ensure completeness before analysis. The validation framework required five essential sections: literature review, methodology, experimental design, dataset specification, and results. Each section needed substantive content exceeding minimal thresholds. Papers failing validation were excluded from subsequent processing. Validated papers were analyzed using GPT-4o-mini model through the OpenAI API. The analysis extracted structured information including research objectives, identified gaps, contributions, and methodological details. Each section underwent independent extraction with temperature parameter 0.3 to ensure consistent results. The system generated cross-pollination research ideas by combining innovations from multiple papers, using temperature 0.7 for creative synthesis.

The implementation relied on the following tools: `arxiv` for API access, `PyPDF2` for PDF text extraction, `requests` for HTTP operations, `openai` for GPT integration, and standard Python libraries (`pathlib`, `json`, `csv`) for data management. The system required an OpenAI API key for content analysis but no authentication for ArXiv access.

## A.6 SECTION-SPECIFIC PERFORMANCE ANALYSIS

**Literature Review Extraction:** The automated system demonstrated moderate accuracy in extracting literature review content, with an average of 8-15 discrepancies per paper. As shown in the heatmap, papers with extensive literature reviews (*Dependent Multiplicities*, *Filter Equivariant Functions*) exhibited correspondingly lower performance scores in this section. Primary challenges include author attribution errors and study count inconsistencies.

**Results Section Processing:** This section exhibited the highest error rates, averaging 10-20 discrepancies per paper, as clearly visible in the heatmap's results column. The automated system struggled particularly with extracting precise numerical results, statistical significance values, and baseline comparison data. Of the 19 papers analyzed, 16 contained placeholder statistical values in the automated version that were marked as "`None`" by human validators.

**Dataset Metadata Extraction:** The heatmap shows relatively consistent performance across papers for dataset information extraction, with sample size extraction showing systematic inconsistencies. Automated systems often provided estimated or approximate values while human validators marked uncertain quantities as missing. This pattern was observed in 12 of 19 papers.

## A.7 AUTOMATED EVALUATION OF GENERATED RESEARCH IDEAS

We evaluate the quality and novelty of generated research ideas through systematic comparison with existing literature using semantic similarity analysis. The evaluation pipeline processes each generated idea through multiple stages to quantify its relationship with published work, enabling scalable assessment across thousands of candidates.

**Idea Summarization and Query Generation.** Each research idea undergoes compression into a searchable summary using GPT-4o-mini with temperature 0.7. The summarization prompt extracts core methodological concepts while removing domain-specific terminology, producing queries of 5-10 words that emphasize generic method names over specific implementations (e.g., "transformer" rather than "BERT"). This abstraction improves retrieval coverage across related work that may use different terminology for similar concepts.

**Literature Retrieval.** We query the OpenAlex database, which indexes over 250 million scholarly works, to retrieve up to 200 relevant papers per idea. The search uses semantic matching on titles and abstracts, filtered to include only articles with available abstracts. To prevent artificial similarity inflation, we exclude source papers used in the idea generation process through title matching, ensuring the evaluation measures genuine similarity to independent work.

**Similarity Computation.** Text embeddings are generated using OpenAI's text-embedding-3-small model (dimension 1536) for both idea summaries and retrieved paper abstracts. We compute cosine similarity between the idea embedding and each paper embedding, producing a distribution of similarity scores. Four aggregate metrics capture different aspects of this distribution: maximum similarity (closest existing work), average similarity (general alignment), top-3 average (robust to outliers), and median similarity (central tendency).

| Similarity Range | Category | Interpretation |
|---|---|---|
| $[0.0, 0.3)$ | Very Novel | Potentially groundbreaking or out-of-scope |
| $[0.3, 0.5)$ | Novel | Significant departure from existing work |
| $[0.5, 0.7)$ | Moderate | Incremental innovation with precedent |
| $[0.7, 0.9)$ | Similar | Strong alignment with existing research |
| $[0.9, 1.0]$ | Very Similar | Potential rediscovery or minor variation |

Table 13: Similarity thresholds for automated research idea evaluation

**Evaluation Metrics.** Ideas are categorized based on maximum similarity scores: high similarity ($\geq 0.7$) indicates strong alignment with existing work, moderate similarity ($0.5 - 0.7$) suggests incremental innovation, and low similarity ($< 0.5$) represents potential novel directions. We track unique paper matching rates to assess whether similar ideas draw from concentrated or diverse literature contexts. Table 13 summarizes the evaluation thresholds and their interpretations.

The evaluation process runs iteratively across all generated ideas, with progress saved every 5 iterations to enable interruption recovery. Each idea's evaluation produces structured output containing similarity metrics, matched papers, and individual paper similarities, enabling both aggregate analysis and detailed inspection of specific idea-literature relationships.

### A.8 EXTRACTION PROCEDURES

#### A.8.1 RESEARCH OBJECTIVES

Extracts the core research problem as a formal statement by identifying the primary question, hypothesis, or challenge addressed in the paper. Transforms informal objectives into structured problems with measurable targets, formatted as: "Given [context], solve [objective] such that [success criteria]."

> **Research Tasks as Problem Statements**
>
> **Purpose:** Extract the core research objective as a formal problem statement by identifying the primary question, hypothesis, or challenge the paper addresses. This includes parsing abstract declarations, introduction statements, and explicit research questions to capture the fundamental intent.
> **Rationale:** Research tasks define the precise boundaries of what the paper seeks to solve. By formalizing these as problem statements rather than vague goals, we create measurable targets against which contributions can be evaluated. This transformation from informal objectives to structured problems enables systematic assessment of whether the research achieves its stated aims.
> **Output:** Action-oriented problem statement following the format: "Given [context/constraints], solve/determine/optimize [specific objective] such that [success criteria]." Include primary and secondary objectives with clear success metrics.
> **Extraction Sources:** Abstract objectives, introduction problem statements, explicit research questions, contribution summaries, conclusion restatements.

#### A.8.2 IDENTIFIED GAPS

Systematically enumerates limitations in prior work across four dimensions: theoretical (conceptual deficiencies), methodological (approach limitations), empirical (evaluation shortcomings), and practical (deployment barriers). Each gap is classified by severity to establish the research motivation.

> **Gaps as Limitation Lists**
>
> **Purpose:** Extract research gaps as structured limitation lists from prior work analysis, distinguishing between theoretical limitations (conceptual gaps), methodological limitations (ap-

proach deficiencies), empirical limitations (evaluation gaps), and practical limitations (deployment challenges).

**Rationale:** Gaps represent concrete deficiencies in existing knowledge that motivate and justify new research directions. Systematic enumeration of these limitations reveals the specific problem space the paper addresses and establishes the novelty claim. Each gap should correspond to a potential contribution, creating a clear motivation-solution narrative.

**Output:** Hierarchically organized limitation list with categories:

- **Theoretical Gaps:** Missing concepts, incomplete frameworks, unproven assumptions

- **Methodological Gaps:** Algorithmic limitations, scalability issues, accuracy problems

- **Empirical Gaps:** Limited evaluation scenarios, missing benchmarks, incomplete comparisons

- **Practical Gaps:** Implementation challenges, deployment barriers, usability issues

**Gap Severity Indicators:** Critical (blocking progress), Major (significant limitation), Minor (incremental improvement opportunity).

### A.8.3 CONTRIBUTIONS

Maps each contribution as an explicit solution to identified gaps, establishing clear correspondence between limitations and innovations. Distinguishes primary contributions from supporting work and documents validation methods and impact assessments for each claimed advancement.

---

**Contributions as Gap-Addressing Solutions**

**Purpose:** Map each contribution as an explicit solution to identified gaps, establishing clear gap-contribution correspondence. Extract both claimed contributions and their validation evidence, distinguishing between primary innovations and supporting contributions.

**Rationale:** Valid contributions must address specific limitations in prior work. This direct mapping ensures research provides targeted solutions rather than arbitrary innovations. The alignment between gaps and contributions validates the research narrative and demonstrates systematic problem-solving rather than opportunistic development.

**Output:** Structured gap-to-solution mappings:

- **Gap Reference:** [Gap ID from limitation list]

- **Contribution Type:** Theoretical/Methodological/Empirical/System

- **Solution Description:** Specific approach addressing the gap

- **Validation Method:** How the contribution is validated

- **Impact Assessment:** Quantitative/qualitative improvement claims

- **Scope:** Conditions under which the solution applies

**Contribution Ranking:** Primary (novel core contributions), Secondary (supporting innovations), Tertiary (implementation details).

---

### A.8.4 METHODOLOGICAL FRAMEWORKS

Decomposes methods into quadruplet specifications: Input (data requirements and constraints), Method (algorithmic procedures and transformations), Output (result formats and metrics), and Detail (implementation parameters and complexity). This structure ensures reproducibility through complete technical specification.

---

### Methods as Input-Method-Output-Detail Quadruplets

**Purpose:** Decompose methodological frameworks into precise four-component specifications that capture the complete computational pipeline. Extract algorithmic procedures, mathematical formulations, system architectures, and implementation strategies with full technical detail.

**Rationale:** Reproducible research requires complete methodological specification beyond high-level descriptions. The quadruplet structure ensures no critical information is omitted: inputs define prerequisites, methods specify transformations, outputs characterize results, and details provide implementation guidance. This systematic decomposition transforms vague methodology sections into actionable specifications.

**Output:** Comprehensive quadruplet tuples for each method component:

- **Input:** Data types, formats, preprocessing requirements, assumptions, constraints
- **Method:** Algorithm steps, mathematical operations, model architectures, optimization procedures
- **Output:** Result formats, post-processing, evaluation metrics, success criteria
- **Detail:** Hyperparameters, convergence criteria, computational complexity, implementation libraries, hardware requirements

**Method Categories:** Core algorithms, preprocessing pipelines, training procedures, inference processes, evaluation protocols.

**Complexity Annotations:** Time complexity, space complexity, sample complexity, communication complexity (if distributed).

---

### A.8.5 EXPERIMENTAL CONFIGURATIONS

Captures comprehensive experimental setup including dataset specifications, baseline configurations, hyperparameter settings, evaluation protocols, and computational resources. Documents statistical validation procedures and ablation studies to enable independent verification and fair comparison.

---

### Complete Experimental Setup

**Purpose:** Document comprehensive experimental configurations capturing every detail necessary for reproduction, including dataset specifications, baseline implementations, evaluation protocols, hyperparameter settings, computational environments, and statistical validation procedures.

**Rationale:** Experimental validity depends on complete configuration disclosure. Missing details prevent replication, while incomplete specifications enable cherry-picking. Full documentation ensures results can be independently verified and fairly compared. Configuration completeness distinguishes rigorous empirical research from anecdotal evidence.

**Output:** Exhaustive configuration specification:

- **Dataset Details:** Sources, sizes, splits, preprocessing, augmentation strategies
- **Baseline Systems:** Versions, configurations, implementation sources, modifications
- **Hyperparameters:** All tunable parameters with search ranges and selection criteria
- **Training Protocol:** Epochs, batch sizes, learning schedules, early stopping criteria
- **Evaluation Metrics:** Primary and secondary metrics with statistical significance tests
- **Computational Resources:** Hardware specs (GPU/CPU/memory), software versions, random seeds
- **Ablation Studies:** Component variations tested with justification
- **Statistical Validation:** Confidence intervals, significance tests, multiple runs

**Reproducibility Checklist:** Code availability, data accessibility, environment specification, result tables with standard deviations.

### A.8.6 Structured Literature Analysis

Transforms unstructured literature reviews into hierarchical taxonomies organized by approach categories, chronological evolution, and theoretical foundations. Identifies research trajectories, convergent trends, and unexplored directions while mapping citation networks and influence patterns.

---

**Organized Prior Work Taxonomy**

**Purpose:** Transform unstructured literature reviews into hierarchical taxonomies that organize prior work by approach categories, chronological evolution, theoretical foundations, and identified limitations. Extract citation relationships, influence patterns, and research lineages.

**Rationale:** Structured literature analysis reveals research trajectories, identifies convergent trends, and positions current work within the broader knowledge landscape. Taxonomic organization exposes patterns in approach evolution, recurring challenges, and unexplored directions. This systematic view distinguishes incremental advances from paradigm shifts.

**Output:** Multi-dimensional literature taxonomy:

- **Approach Categories:**
    - Classical methods with key papers and limitations
    - Modern approaches with innovations and trade-offs
    - Emerging directions with potential and challenges
- **Evolution Timeline:** Chronological development with breakthrough papers
- **Theoretical Foundations:** Underlying principles, assumptions, mathematical frameworks
- **Performance Landscape:** Comparative analysis of approach capabilities
- **Open Problems:** Unresolved challenges identified across multiple works
- **Citation Network:** Key papers, influence relationships, research clusters

**Analysis Dimensions:** Scalability progression, accuracy improvements, complexity evolution, application domain expansion.

**Gap Synthesis:** Cross-cutting limitations that motivate current research direction.

---

### A.9 An Example on Idea Procedure

This example demonstrates the application of the methodological framework extraction process to a real research paper. We analyze "Swin Transformer V2: Scaling Up Capacity and Resolution" to illustrate how complex methodological contributions can be systematically decomposed into structured procedural representations. The Swin Transformer V2 paper presents multiple technical innovations for scaling vision transformers to larger capacities and resolutions. By applying our extraction framework, we identify six distinct procedures that constitute the paper's methodological contributions. Each procedure is captured as an input-method-output triplet with accompanying technical details, forming the quadruplet structure described in our framework. It reveals how the paper addresses key challenges in vision transformer scaling.

```json
{
  "procedures": [
    {
      "procedure_name": "Residual Post-Norm",
      "triplet": {
        "input": [
          "activation values from residual blocks"
        ],
        "method": [
          "apply layer normalization after merging residual output
              with main branch"
        ],
        "output": [
          "milder activation values across layers"
```

```
14            ]
15          },
16          "technical_details": {
17            "parameters": {
18              "layer_norm_position": "post"
19            },
20            "algorithms": [
21              "Layer Normalization"
22            ],
23            "tools": []
24          }
25        },
26        {
27          "procedure_name": "Scaled Cosine Attention",
28          "triplet": {
29            "input": [
30              "query and key vectors"
31            ],
32            "method": [
33              "compute attention logits using scaled cosine function"
34            ],
35            "output": [
36              "normalized attention values"
37            ]
38          },
39          "technical_details": {
40            "parameters": {
41              "scaling_factor": "greater than 0.01"
42            },
43            "algorithms": [
44              "Cosine Similarity"
45            ],
46            "tools": []
47          }
48        },
49        {
50          "procedure_name": "Log-Spaced Continuous Position Bias
                  (Log-CPB)",
51          "triplet": {
52            "input": [
53              "log-spaced coordinates"
54            ],
55            "method": [
56              "use a meta network to generate bias values for arbitrary
                    coordinates"
57            ],
58            "output": [
59              "continuous position bias values for varying window sizes"
60            ]
61          },
62          "technical_details": {
63            "parameters": {
64              "meta_network_layers": 2,
65              "activation_function": "ReLU"
66            },
67            "algorithms": [
68              "Meta Network"
69            ],
70            "tools": []
71          }
72        },
73        {
74          "procedure_name": "Self-Supervised Pre-training (SimMIM)",
75          "triplet": {
76            "input": [
```

```
 77              "unlabeled images"
 78            ],
 79            "method": [
 80              "train model using masked image modeling"
 81            ],
 82            "output": [
 83              "pre-trained model with reduced dependency on labeled data"
 84            ]
 85          },
 86          "technical_details": {
 87            "parameters": {
 88              "dataset_size": "70 million labeled images"
 89            },
 90            "algorithms": [
 91              "Masked Image Modeling"
 92            ],
 93            "tools": []
 94          }
 95        },
 96        {
 97          "procedure_name": "GPU Memory Optimization",
 98          "triplet": {
 99            "input": [
100              "large model parameters"
101            ],
102            "method": [
103              "apply Zero-Redundancy Optimizer, activation checkpointing,
                     and sequential self-attention computation"
104            ],
105            "output": [
106              "reduced GPU memory consumption"
107            ]
108          },
109          "technical_details": {
110            "parameters": {
111              "optimizer": "ZeRO stage-1",
112              "activation_checkpointing": "enabled",
113              "self_attention_computation": "sequential"
114            },
115            "algorithms": [
116              "Zero-Redundancy Optimizer",
117              "Activation Checkpointing"
118            ],
119            "tools": [
120              "DeepSpeed"
121            ]
122          }
123        },
124        {
125          "procedure_name": "Model Configuration",
126          "triplet": {
127            "input": [
128              "model architecture specifications"
129            ],
130            "method": [
131              "define stages, blocks, and channels for different model
                     sizes"
132            ],
133            "output": [
134              "configured model for training"
135            ]
136          },
137          "technical_details": {
138            "parameters": {
139              "model_sizes": [
```

```
140        "SwinV2-T",
141        "SwinV2-S",
142        "SwinV2-B",
143        "SwinV2-H",
144        "SwinV2-G"
145      ],
146      "channels": [
147        96,
148        128,
149        192,
150        352,
151        512
152      ],
153      "blocks": [
154        [2, 2, 6, 2],
155        [2, 2, 18, 2],
156        [2, 2, 18, 2],
157        [2, 2, 42, 4]
158      ]
159    },
160    "algorithms": [],
161    "tools": []
162    }
163    }
164  ]
165 }
```

Listing 1: Idea Procedures Extracted from "Swin Transformer V2: Scaling Up Capacity and Resolution"

## A.10 IDEA GENERATION CROSS TASK WITHIN FIELD

Table 14 presents similarity proportions for four distinct topics (CV, DC, CL, LG) across three different combinations (1:1, 3:1, and 40:0). The results indicate considerable variability depending on both the combination and the specific topic. CV and LG show relatively higher similarity rates, peaking at 4.88% (CV 1:1) and 4.25% (LG 40:0), respectively, suggesting these domains are more prone to idea overlap when closely matched resources or extensive paper bases are used. In contrast, CL consistently exhibits lower similarity rates ranging from 1.12% to 2.25%, implying less overlap and possibly greater originality. Additionally, the relatively low overall similarity percentages suggest that cross-task idea generation, where ideas from one task substantially overlap with those from another distinct task, is comparatively less common.

| Domain | 1:1 | 3:1 | 40:0 |
|---|---|---|---|
| Computer Vision Task Combination | 4.88 | 2.62 | 3.88 |
| Data Science Task Combination | 3.83 | 3.13 | 2.25 |
| Computational Linguistics Task Combination | 1.12 | 2.25 | 2.12 |
| Machine Learning Task Combination | 3.34 | 2.71 | 4.25 |

Table 14: High Similarity Percentages Across Different Runs

## A.11 COMPUTATIONAL COST ANALYSIS

We evaluated the computational costs of three experimental configurations to assess the economic feasibility of large-scale research idea generation. The baseline system uses GPT-4o with chain-of-ideation prompting to process raw paper text directly. The enhanced system employs GPT-4o-mini with structured paper representation extraction followed by CoI-based generation, while the full system implements our complete pipeline using GPT-4o-mini for both preprocessing and cross-pollination generation. Cost calculations are based on current API pricing of $5.00 and $15.00 per

million input and output tokens respectively for GPT-4o, compared to $0.15 and $0.60 for GPT-4o-mini.

| System | Model | Input Tokens | Output Tokens | Total Tokens | Cost ($) |
|--------|-------|-------------|--------------|-------------|---------|
| Baseline (CoI) | GPT-4o | 3,350 | 1,950 | 5,300 | 0.0460 |
| Enhanced System | GPT-4o-mini | 11,950 | 5,850 | 17,800 | 0.0053 |
| Full System | GPT-4o-mini | 17,100 | 8,200 | 25,300 | 0.0075 |

Table 15: Token usage and cost per generated research idea

Table 15 presents the token consumption breakdown per generated idea. The baseline system requires 3,350 input and 1,950 output tokens across its brainstorming, JSON formatting, and instruction generation phases, resulting in a cost of $0.046 per idea. In contrast, the enhanced system processes two papers requiring 10,300 input and 4,700 output tokens for preprocessing, plus 1,650 input and 1,150 output tokens for generation, totaling $0.0053 per idea. The full system processes three papers with correspondingly higher preprocessing costs of 15,450 input and 7,050 output tokens, maintaining the same generation overhead for a total cost of $0.0075 per idea.

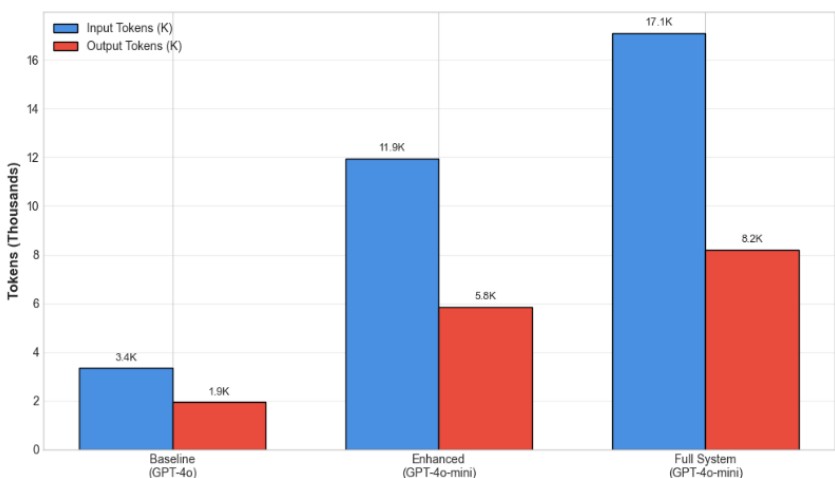

Figure 7: Cost comparison across three configurations

Figure 7 illustrates that despite processing significantly more tokens, the enhanced and full systems achieve cost reductions of 88.5% and 83.7% respectively compared to the baseline. This counterintuitive result stems from the 30-fold reduction in per-token costs when using GPT-4o-mini versus GPT-4o, which more than compensates for the additional preprocessing overhead. The cost differential becomes particularly pronounced at scale, where generating 10,000 ideas would cost $460 with the baseline system compared to only $53 with the enhanced system and $75 with the full system, representing reduction factors of 8.68× and 6.13× respectively.

The analysis reveals that structured paper representations enable smaller language models to achieve comparable performance to larger models on specialized tasks. While paper preprocessing requires approximately 7,500 tokens per paper, this upfront investment amortizes across multiple idea generation tasks using the same corpus and enables more efficient downstream processing. These findings demonstrate that our approach successfully addresses the computational cost barrier to large-scale automated research ideation, which makes it economically feasible for broader deployment in academic and industrial settings while maintaining quality standards. The substantial cost reductions achieved without compromising output quality represent a significant advancement toward democratizing AI-assisted research tools.

