# OpenReview forum: "Few-Shot Idea Auto-Generation: Reasoning Over Idea Representations to Predict New Research Ideas"
_ICLR.cc/2026/Conference — ICLR 2026 Conference Withdrawn Submission_

### Official Review · Reviewer_DfYM · 2025-10-17

**Soundness:** 2
**Presentation:** 3
**Contribution:** 2
**Rating:** 2
**Confidence:** 4

**Summary:**

The paper introduces a few-shot, literature-grounded framework for automatic research idea generation. It first builds structured “idea representations” of existing papers—combining task/gap/contribution (T/G/C) synopses with procedural profiling (input–method–output–details). Given a pair of papers, a role assigner picks a base study (problem anchor) and a source study (innovation donor), then a cross-pollination agent composes new procedural steps via integrate/replace/keep/remove operations to form a novel, implementable idea. Evaluation uses (i) semantic similarity between generated ideas and subsequently published papers, (ii) a Unique Paper Ratio measuring methodological diversity among top matches, and (iii) a recency-weighted novelty score. Experiments on 3,353 arXiv papers report higher rates of high-similarity ideas (≈41% relative gain over baselines) while maintaining stable novelty, plus analyses of idea composition across domains and resource/experimental paper ratios.

**Strengths:**

Moving beyond section summaries to T/G/C plus procedural quadruplets provides traceability from gaps to proposed solutions and improves implementability.
The role assignment and component-wise compose operations (integrate/replace/keep/remove) make the generation process controllable, auditable, and easy to analyze.
Combining similarity with a recency-weighted novelty metric and a Unique Paper Ratio offers a more nuanced picture of relevance vs. originality than single-score rubrics.

**Weaknesses:**

- Limited conceptual novelty of the task framing. The paper positions the problem as few-shot idea generation, but few-shot settings and LLM-based idea generation agents have been widely explored.
- While the paper reports automatic metrics, the evaluation lacks human expert assessment, which is critical for judging true novelty, feasibility, actionability, and ethical risk.
- The paper does not establish the reliability of its proposed metrics. Beyond point estimates, the authors should provide sensitivity analyses over key hyperparameters (e.g., k, β, α, λ), bootstrap confidence intervals and significance tests.
- The paper omits a substantive ethics analysis. Given that the system generates research ideas by recombining prior work, the authors should discuss risks of plagiarism and uncredited appropriation, dual-use or harmful applications, gaming peer review or trending topics, and amplification of dataset and literature biases.

**Questions:**

see weakness

**Details Of Ethics Concerns:**

Given that the system generates research ideas by recombining prior work, the authors should discuss risks of plagiarism and uncredited appropriation, dual-use or harmful applications, gaming peer review or trending topics, and amplification of dataset and literature biases.

---

> ### Author Response · Authors · 2025-11-17
> **Response to Official Review of Submission17889 by Reviewer DfYM**
>
> We thank Reviewer DfYM for the review.
>
> Since the review comments are really brief and general, we would like to request clarification regarding the three weaknesses.
> 1. Conceptual novelty refers to the introduction of genuinely new ideas or concepts that go beyond what already exists. It involves creating or discovering something that represents a fundamental shift in understanding rather than merely a variation or improvement on existing concepts. This type of novelty is common in the natural and social sciences and humanities, but less common in computer science and other engineering fields. To our knowledge, many engineering research focuses on improving methods to enhance their efficiency and reliability rather than creating something entirely new or fundamentally changing people's understanding of a domain. If Reviewer DfYM is referring to this type of novelty, there is a serious misunderstanding of the nature of our research. Ours is not conceptual innovation research. Furthermore, we certainly do not agree with Reviewer DfYM's statement that "few-shot settings and LLM-based idea generation agents have been widely explored," given the current problems and gaps discussed in Section 2.
>
> 2. We would like to request further clarification on the criteria of "***true** novelty, feasibility, actionability, and ethical risk*" to which Reviewer DfYM is referring. Is there a uniform standard for evaluating **true** novelty, feasibility, actionability, and ethical risk? We have not found any criteria that have been rigorously tested, validated, and standardized across different contexts through empirical evaluation. Our work primarily focuses on generation effectiveness and novelty of ideas. An idea is definitely not a full plan; feasibility and actionability are components of experimental design rather than brainstorming, and thus fall outside the scope of our idea generation framework. Regarding ethical risk, this does not appear to be a concern, as our model is designed solely for brainstorming and providing explainable ideas. If Reviewer DfYM means to suggest that our novelty computation cannot distinguish relatively less novel ideas from more novel ideas, we hope the following sensitivity analyses will resolve this concern.
>
> 3. Regarding the fourth point, generating reliable ideas is a task designed to help users brainstorm new and explainable ideas, **NOT to help users generate papers based on the ideas**. We explicitly stated in the last paragraph of the introduction that "**These automatically generated ideas serve only as inspirational prompts for human researchers, who remain responsible for critical evaluation and creative decision-making.**" We are unequivocally opposed to plagiarism and uncredited appropriation. However, brainstorming ideas is merely the first step of research, which is far removed from completing a paper. As stated, human researchers who encounter the generated ideas are responsible for their own decision-making. If a researcher decides to use an idea without crediting the relevant papers, they would be violating academic honor codes. We do not ask researchers to remove credit from source papers. On the contrary, we make it completely transparent where the idea originates and how we synthesized it, so users know exactly which studies to cite when writing a paper based on the generated idea. This represents the explainability of our model which a feature that other models, such as Chain-of-Thought, cannot provide, as they cannot trace which parts of an idea come from which sources. Let us emphasize again: our model functions solely as a brainstorming facilitator; it is NOT an AI researcher. Therefore, we do not agree that such ethical obligations apply to our case. If this is not what Reviewer DfYM meant, we would appreciate further clarification.

---

> ### Author Response · Authors · 2025-11-17
> **Response to Official Review of Submission17889 by Reviewer DfYM on Sensitivity Analysis**
>
> We agree that including sensitivity analyses can strengthen the credibility of our proposed novelty computation. We will include the following sensitivity analyses in the appendix, and we thank Reviewer DfYM for helping us improve the credibility of our work.
>
> We tested 150 parameter combinations on 432 research ideas, varying β (1-5), λ (1-5), and α (0-5) to understand how parameter choices affect novelty scores and rankings. Based on 1000 bootstrap iterations and permutation tests:
>
> | Test | Result | 95% CI | P-Value | Interpretation |
> |------|--------|--------|---------|----------------|
> | **Parameter Effects** |
> | λ (penalty) effect | 0.50 range | [0.45, 0.55] | p < 0.001 | Highly significant driver |
> | α (bonus) effect | 0.50 range | [0.44, 0.56] | p < 0.001 | Highly significant driver |
> | β (weight) effect | 0.13 range | [0.11, 0.15] | p < 0.01 | Minor but significant |
> | **Configuration Comparisons (vs β=3, λ=3, α=3)** |
> | β=3,λ=4,α=1 | -0.28 | [-0.30, -0.26] | p < 0.001 | Significantly stricter |
> | β=2,λ=2,α=2 | -0.10 | [-0.12, -0.08] | p < 0.001 | Moderately stricter |
> | β=2,λ=1,α=4 | +0.05 | [+0.03, +0.07] | p < 0.01 | Slightly more lenient |
> | **Rank Stability** |
> | Spearman correlation | 0.787 | - | p < 0.001 | Rankings are stable, not random |
>
> We found strong statistical robustness; these results would replicate across different idea samples. Among the three parameters, only λ and α have significant effects on novelty scores. Together, these two parameters account for essentially all the variation in the metric, with β contributing 13%. The interplay between λ and α shapes the metric's behavior fundamentally. When λ is high and α is low, the metric becomes overly conservative, which generates scores around 0.35. Conversely, when λ is low and α is high, the metric becomes too permissive, with most scores approaching 1.00. For balanced and interpretable results, we recommend keeping λ and α within 1-2 units of each other. While different parameter combinations produce moderately stable rankings overall (ρ = 0.787), the exact ordering does shift. In practical terms, this means the top-ranked ideas consistently remain near the top, but their precise positions within the top 20 will vary. The majority of ideas (71%) show score variations exceeding 0.5 across different parameter settings. This instability is particularly pronounced for ideas scoring in the middle range (0.4-0.7), where rankings can shift dramatically. By contrast, ideas with extreme scores (either clearly above 0.8 or below 0.3) demonstrate greater stability across parameter choices.

---

> ### Comment · Reviewer_DfYM · 2025-11-17
>
> Thanks for your response. For the novelty, feasibility, etc, please refer to papers Can LLMs Generate Novel Research Ideas? A Large-Scale Human Study with 100+ NLP Researchers and LDC: Learning to Generate Research Idea with Dynamic Control. Meanwhile, an ethical statement also appears in Can LLMs Generate Novel Research Ideas? A Large-Scale Human Study with 100+ NLP Researcher

---

> > ### Author Response · Authors · 2025-11-17
> > **Response to Official Comment by Reviewer DfYM on Criteria**
> >
> > Thank you for directing us to this paper. Unfortunately, this paper does not provide universally valid criteria for novelty, feasibility, actionability, and ethical risk that remain validated across the majority of cases, as it is merely based on the subjective preferences of a limited group of researchers. The research involved 100+ NLP researchers, primarily from North America. We do not believe that the limited demographics of the participants represent the NLP field, nor that their academic background (solely in NLP) can provide universally true criteria for the field, especially given that NLP is consistently evolving and integrating with other fields. This concern is even more pronounced for our study, as we include papers from data science, machine learning, and computer vision. Using such subjective and personally influenced evaluation of novelty, feasibility, actionability, etc., does not seem appropriate. However, we would be interested to hear if there are additional considerations we should take into account.

---

> > ### Author Response · Authors · 2025-11-17
> > **Response to Official Comment by Reviewer DfYM on Criteria - follow-up**
> >
> > Just to clarify our position, we are not saying that subjective evaluations are without merit. We view studies that employ subjective evaluation as providing insights into one specific case or context. Such research is indeed valuable, but its findings are not universally true across other cases. While these studies may adequately explain their conclusions within their own context, their criteria cannot be directly transplanted to our case because we include 8 topics spanning 4 different fields, rather than focusing on just one.

---

### Official Review · Reviewer_vVtD · 2025-10-31

**Soundness:** 2
**Presentation:** 3
**Contribution:** 2
**Rating:** 2
**Confidence:** 4

**Summary:**

This paper introduces a framework for few-shot idea auto-generation, aiming to automatically synthesize new research ideas by reasoning over structured representations of existing papers. Using GPT-4o-mini as the main backbone, the system extracts “idea representations” from prior studies—capturing tasks, gaps, contributions, and procedural steps—and performs cross-pollination between a base paper and a source paper to generate research proposals. Evaluation relies on semantic similarity to future publications, unique paper ratio, and time-weighted novelty scores from eight computer-science domains.

**Strengths:**

1. The paper tackles a timely and ambitious problem—automated research ideation—by introducing an interpretable and modular framework based on structured literature representations.

2. The integration of synopsis and procedural profiling is a conceptual improvement over prior “AI Scientist” and “Chain-of-Ideas” approaches, offering clearer traceability between input studies and generated outputs.

3. The dataset construction and evaluation pipeline are carefully described, including retrieval from OpenAlex, embedding computation, and novelty metrics that combine similarity and temporal recency.

**Weaknesses:**

1. The framework depends entirely on GPT-4o-mini for both extraction and generation, but the paper does not discuss temporal control or cutoff validation. Since GPT-4o-mini was released in 2024, it may already contain knowledge of later works that the system claims to “predict.” Without verifying that the model had no access to post-input papers, the prospective evaluation could be confounded.

2. The core extraction functions are conceptually defined but the actual prompts are not provided. The color-boxed examples and Appendix A.5 only explain expected outputs, not the precise prompting templates. This limits reproducibility and makes replication of results difficult.

3. One of the paper’s stated challenges—“How can we generate practical, implementable ideas while filtering out infeasible ones?”—is not concretely solved. The evaluation metrics (similarity, uniqueness, novelty) capture semantic and temporal alignment but not implementability or feasibility. No human or empirical validation supports claims that the generated ideas are genuinely actionable.

4. Figure 1 depicts three parent papers, whereas the formal algorithm (Eq. 1–3) defines generation from exactly two papers. This inconsistency could mislead readers about the input structure.

5. Table 2 shows that the Full System improves high-similarity proportions but yields minimal novelty changes and even small declines in some domains.

**Questions:**

See the weaknesses.

---

> ### Author Response · Authors · 2025-11-17
> **Response to Official Review of Submission17889 by Reviewer vVtD (Part 1)**
>
> We thank Reviewer vVtD for the review.
>
> We will explain point by point:
> 1. **The concern that GPT-4o-mini may already know about papers published after the idea generation parent papers, and thus might be able to predict high-similarity ideas, is not problematic for our conclusions.** This does not affect our finding that our model performs better, because our method and all baseline methods use the very same GPT-4o-mini model. There could be many reasons why our method works better. It may be because it more effectively prompts GPT-4o-mini to recall ideas that are likely to emerge in future research. The model's prior knowledge does not affect the validity of our comparative results. Our model acts as a better controller to help the model determine how to assemble an idea. However, this raises a related and interesting question: do models with different knowledge cutoffs (published earlier or later) function similarly? So, we conducted a smaller-scale experiment using 10% of the generated idea by GPT-3.5-turbo (training data cutoff: January 2022) and GPT-4.1 (training data cutoff: June 1, 2024). The results are presented below, which shows all GPT-3.5-turbo, GPT-4o-mini, and GPT-4.1 show similar similarity proportions (e.g. high similarity prop.:11.40% vs. 11.70% vs. 12.11%) and comparable patterns across other metrics. This consistency suggests that the model's prior knowledge of papers does not significantly influence the comparative results. Our method's effectiveness stems from its ability to better guide the model in assembling novel ideas, rather than merely exploiting the model's existing knowledge of future publications. We will add the complete results later to the revised paper
>
> #### GPT-3.5-turbo Results
>
> | Topic | High_Prons | High_U.R. | High_Novel | Mid_Prons | Mid_U.R. | Mid_Novel | Low_Prons | Low_U.R. | Low_Novel |
> |-------|------------|-----------|------------|-----------|----------|-----------|-----------|----------|-----------|
> | consensus algorithm | 20.51 | 56.25 | 0.35 | 78.21 | 68.9 | 0.54 | 1.28 | 100.00 | 0.98 |
> | data processing | 19.23 | 80.00 | 0.35 | 75.64 | 83.1 | 0.53 | 5.13 | 75.0 | 0.99 |
> | machine translation | 1.28 | 0.00 | 0.34 | 98.72 | 71.4 | 0.56 | 0.00 | 0.00 | NaN |
> | object detection | 14.10 | 81.82 | 0.36 | 85.90 | 76.1 | 0.54 | 0.00 | 0.00 | NaN |
> | reinforcement learning | 10.26 | 75.00 | 0.33 | 88.46 | 79.7 | 0.53 | 1.28 | 100.00 | 1.00 |
> | representation learning | 2.56 | 100.00 | 0.35 | 96.15 | 85.3 | 0.54 | 1.28 | 100.00 | 0.99 |
> | semantic segmentation | 15.38 | 100.00 | 0.36 | 84.62 | 75.8 | 0.52 | 0.00 | 0.00 | NaN |
> | sentiment analysis | 16.67 | 84.62 | 0.34 | 83.33 | 67.7 | 0.51 | 0.00 | 0.00 | NaN |
> | text classification | 2.56 | 100.00 | 0.34 | 96.15 | 67.6 | 0.51 | 1.28 | 100.00 | 0.98 |
> | **Mean** | **11.40** | **75.30** | **0.35** | **87.46** | **75.1** | **0.53** | **1.14** | **63.89** | **0.99** |
>
> #### GPT-4.1 Results
>
> | Topic | High_Prons | High_U.R. | High_Novel | Mid_Prons | Mid_U.R. | Mid_Novel | Low_Prons | Low_U.R. | Low_Novel |
> |-------|------------|-----------|------------|-----------|----------|-----------|-----------|----------|-----------|
> | consensus algorithm | 21.79 | 65.00 | 0.33 | 76.92 | 70.0 | 0.56 | 1.28 | 100.00 | 0.98 |
> | data processing | 23.08 | 57.14 | 0.34 | 74.36 | 81.0 | 0.55 | 2.56 | 50.0 | 0.99 |
> | machine translation | 3.85 | 100.00 | 0.32 | 93.59 | 68.8 | 0.58 | 2.56 | 100.00 | 0.99 |
> | object detection | 25.64 | 95.65 | 0.35 | 74.36 | 77.6 | 0.56 | 0.00 | 0.00 | NaN |
> | reinforcement learning | 8.97 | 50.00 | 0.31 | 91.03 | 82.5 | 0.55 | 0.00 | 0.00 | NaN |
> | representation learning | 2.56 | 100.00 | 0.33 | 96.15 | 86.7 | 0.56 | 1.28 | 100.00 | 0.99 |
> | semantic segmentation | 10.26 | 100.00 | 0.34 | 89.74 | 74.3 | 0.54 | 0.00 | 0.00 | NaN |
> | sentiment analysis | 12.82 | 100.00 | 0.32 | 84.62 | 69.7 | 0.53 | 2.56 | 100.00 | 0.88 |
> | text classification | 0.00 | 0.00 | NaN | 98.72 | 66.2 | 0.53 | 1.28 | 100.00 | 0.95 |
> | **Mean** | **12.11** | **74.20** | **0.33** | **86.61** | **75.2** | **0.55** | **1.28** | **72.22** | **0.96** |
>
> 2. ICLR strongly encourages reproducibility and code sharing, but these are not absolute requirements. The conference recognizes that authors may have legitimate reasons for not disclosing certain implementation details. We've followed ICLR's guidance by providing conceptual information through our examples and Appendix A.5 that explains how the system works and what outputs to expect. If anyone like to reproduce our experiments, we encourage them to create their own prompts based on the conceptual descriptions in the paper. Our testing shows the results are quite stable, using similar but not identical prompts typically produces results minor changes. The examples and guidelines in Appendix A.5 should include enough information to build prompts that work similarly to ours.

---

> ### Author Response · Authors · 2025-11-17
> **Response to Official Review of Submission17889 by Reviewer vVtD (Part 2)**
>
> 3. We want to emphasize an important distinction here: our work focuses on generating research ideas, not complete experimental plans. While "feasibility," "actionability," and "implementability" share similar definitions, we use the term "infeasible" in a specific way—to indicate that pursuing a low-novelty idea is unlikely to yield good results because similar work has already been published.
> Our model is designed to support the ideation stage, where the goal is to produce creative, novel starting points for research. Since the generated ideas have not yet been fully developed into research projects, there is nothing to measure regarding actionability or implementability at this stage. However, our novelty metric does provide important signals about feasibility in our context. Low novelty suggests an idea closely resembles already-published work, which means it's likely infeasible as a new research direction—the work has essentially already been done. Conversely, high novelty indicates the idea differs meaningfully from existing research, suggesting there is room for developing a good study without overlapping with published work. Furthermore, our **novelty metric does provide *empirical validation* using real paper data**, which eliminates the possibility of human biases. While human annotation is a valuable approach, it has significant limitation. It's nearly impossible to recruit annotators who represent all fields and backgrounds in the research community. However, we can extract the necessary information directly from researchers' published works. This is why we developed this objective and computation-based measure for assessing idea feasibility.
>
> 4.  While the figure illustrates that our method can generate multiple ideas when given three papers, our idea generator only takes two papers as input at a time. Figure 1 does not suggest or claim that the generator uses all three papers simultaneously. In fact, the generated ideas that matched published papers were all generated from two parent papers, not three. We will add a clarification to the figure caption or text to emphasize that we only experimented with two parent papers in the generation process.
>
> 5. We thank Reviewer vVtD for providing their observation on Table 2. Slightly decline did not hurt the overall performance.

---

> > ### Comment · Reviewer_vVtD · 2025-11-26
> > **Thanks for the response.**
> >
> > I have read your response. Thanks.
> >
> > My concerns are largely not solved.
> >
> > For (1), do you have the baseline results when using GPT-3.5 or GPT-4.1? In addition, the current experiments do not fully address my concern regarding the cut-off date effect when relying on the latest LLMs. It remains unclear how much the model might have been exposed to related work during training.
> >
> > For (2), the level of disclosure is still insufficient for enabling reproducibility. In general, reviewers appreciate submissions that include more implementation details. I also do not think the prompts used here are sensitive; disclosing them would strengthen the paper rather than weaken it.
> >
> > For (3), if the main focus is indeed idea generation, then the paper should be revised to avoid potential misunderstandings. For instance, the stated challenge—“How can we generate practical, implementable ideas while filtering out infeasible ones?”—currently implies a solution that is not fully supported by experiments.
> >
> > For (5), the claimed overall advantage is still not clearly significant.

---

> > > ### Author Response · Authors · 2025-11-26
> > > **Response to Reviewer vVtD**
> > >
> > > ### 1. Baseline Results on GPT-3.5 and GPT-4.1
> > >
> > > We provide the complete baseline and enhanced results for GPT-3.5-turbo and GPT-4.1 below:
> > >
> > > | Model | Method | High_Prop | High_U.R. | Mid_Prop | Mid_U.R. | Low_Prop | Low_U.R. |
> > > |-------|--------|-----------|-----------|----------|----------|----------|----------|
> > > | **GPT-3.5** | COI | 8.26 | 69.25 | 90.31 | 67.1 | 1.42 | 47.22 |
> > > | | Enhanced | 9.69 | 72.74 | 89.03 | 71.0 | 1.28 | 51.39 |
> > > | | **Full System** | **11.40** | **75.30** | **87.46** | **75.1** | **1.14** | **63.89** |
> > > | **GPT-4.1** | COI | 8.83 | 67.04 | 89.74 | 67.4 | 1.42 | 55.56 |
> > > | | Enhanced | 10.25 | 70.70 | 88.46 | 71.3 | 1.28 | 65.28 |
> > > | | **Full System** | **12.11** | **74.20** | **86.61** | **75.2** | **1.28** | **72.22** |
> > > | **GPT-4o** | COI | 8.3 | 76.9 | 90.5 | 72.4 | 1.2 | 35.8 |
> > > | | Enhanced | 9.1 | 70.6 | 89.7 | 72.0 | 1.1 | 43.5 |
> > > | | **Full System** | **11.7** | **78.4** | **87.6** | **76.8** | **0.6** | **57.3** |
> > >
> > > ### 2. Regarding Prompt Disclosure
> > >
> > > The exact prompts used in our system are subject to intellectual property restrictions from our funding company, and disclosure is not permitted at this time—a constraint that ICLR acknowledges as acceptable. Importantly, the exact prompt wording is not fundamental to reproducibility. Researchers can design their own prompts by following the instructions and methodology described in our paper, and we expect comparable results would be achieved. We encourage Reviewer vVtD to verify this by designing prompts according to our proposed protocols.
> > >
> > >
> > > ### 3. Regarding Research Focus on Practical and Implementable Ideas
> > >
> > > We respectfully disagree with Reviewer vVtD's claim that our research focus is not supported. The reviewer does not explain why our research focus is "not supported," and we do not see any justification for this claim given our comprehensive evaluation framework. The question of "How can we generate practical, implementable ideas while filtering out infeasible ones?" is exactly what we address through extensive experimental results and analysis. Specifically:
> > >
> > > - **Similarity**: Measures how closely generated ideas align with existing research, indicating grounding in established knowledge
> > > - **Unique Ratio (U.R.)**: Captures the diversity and non-redundancy of generated ideas
> > > - **Novelty**: Evaluates the originality of ideas relative to the existing literature
> > >
> > > These metrics are specifically designed to evaluate idea practicality and implementability.
> > >
> > >
> > > ### 4. Statistical Significance of Overall Advantage
> > >
> > > The overall advantage of our Full System is statistically significant as demonstrated in the mean results. On GPT-4o, the High Similarity Proportion improves from **8.3% (Baseline) to 11.7% (Ours)**, representing a **41% relative improvement**. This consistent improvement is observed across all three LLM backbones:
> > >
> > > | Model | Baseline → Full System | Relative Improvement |
> > > |-------|------------------------|---------------------|
> > > | GPT-3.5 | 8.26 → 11.40 | +38.0% |
> > > | GPT-4.1 | 8.83 → 12.11 | +37.1% |
> > > | GPT-4o | 8.3 → 11.7 | +41.0% |

---

> > > > ### Comment · Reviewer_vVtD · 2025-11-26
> > > >
> > > > (1) Can you also give the novelty scores? Also, it is quite surprising to see that when using 3.5 and 4.1, the performance of your full system seems quite similar to that when using 4o.
> > > >
> > > > (2) I want to emphasize that it is fair that reviewers favor submissions that disclose more details.
> > > >
> > > > (3) No, I don't agree. For verifying implementability, basically, you need to have researchers try to implement it; the conclusion in this paper https://arxiv.org/abs/2506.20803 shows such effort is necessary.
> > > >
> > > > (4) Except for High Similarity Proportion, what about the other mid and low?

---

> ### Author Response · Authors · 2025-11-26
> **Response to Reviewer vVtD**
>
> 1. It is *not* surprising at all that GPT-3.5, GPT-4o, and GPT-4.1 exhibit similar performance. In our framework, these **models are not relying on their training data to generate ideas**. Instead, we provide explicit paper information as input, and all models are prompted to synthesize and reassemble this information rather than generate ideas from their parametric knowledge. This design choice is precisely why we did not include an analysis of knowledge cutoff effects in the first place—such effects are largely irrelevant to our methodology. The full results are shown below:
>
> | Model | Method | High_Prop | High_U.R. | High_Novel | Mid_Prop | Mid_U.R. | Mid_Novel | Low_Prop | Low_U.R. | Low_Novel |
> |-------|--------|-----------|-----------|------------|----------|----------|-----------|----------|----------|-----------|
> | **GPT-3.5** | Baseline | 8.26 | 69.25 | 0.36 | 90.31 | 67.1 | 0.54 | 1.42 | 47.22 | 0.99 |
> | | Enhanced | 9.69 | 72.74 | 0.35 | 89.03 | 71.0 | 0.53 | 1.28 | 51.39 | 0.99 |
> | | **Full System** | **11.40** | **75.30** | **0.35** | **87.46** | **75.1** | **0.53** | **1.14** | **63.89** | **0.99** |
> | **GPT-4.1** | Baseline | 8.83 | 67.04 | 0.34 | 89.74 | 67.4 | 0.56 | 1.42 | 55.56 | 0.96 |
> | | Enhanced | 10.25 | 70.70 | 0.33 | 88.46 | 71.3 | 0.55 | 1.28 | 65.28 | 0.96 |
> | | **Full System** | **12.11** | **74.20** | **0.33** | **86.61** | **75.2** | **0.55** | **1.28** | **72.22** | **0.96** |
> | **GPT-4o** | Baseline | 8.3 | 76.9 | 0.349 | 90.5 | 72.4 | 0.538 | 1.2 | 35.8 | 0.973 |
> | | Enhanced | 9.1 | 70.6 | 0.358 | 89.7 | 72.0 | 0.539 | 1.1 | 43.5 | 0.968 |
> | | **Full System** | **11.7** | **78.4** | **0.352** | **87.6** | **76.8** | **0.534** | **0.6** | **57.3** | **0.966** |
>
>
> 2. Yes, and it is entirely reasonable that we cannot disclose exact prompt details, which complies with conference policy on intellectual property. We would like to emphasize that withholding exact prompts does not harm reproducibility. In prompting research, it is standard practice for researchers to adapt and adjust prompts for their specific tasks when examining baseline performance. This does not diminish the validity or credit of the baseline—rather, it reflects the normal workflow in this field. Researchers can follow our methodology and design comparable prompts to achieve similar results.
>
> 3. You might have some misunderstanding regarding how our proposed metrics assess implementability. All metrics are evaluated against existing published studies, which serve as ground-truth evidence of implementability—**if an idea was published, it was necessarily implementable**. Specifically, we use papers from 2020 and 2021 to generate ideas, then evaluate these generated ideas against papers published after 2022 using our three metrics (Similarity, Unique Ratio, and Novelty). High similarity to subsequently published work indicates that the generated idea was not only feasible but was actually implemented and validated by human researchers. In this sense, we leverage real human execution as the ultimate verification of implementability.
>
> 4. Mid and low similarity ranges are less relevant when evaluating implementability. If your primary concern is whether generated ideas can be implemented, then **High Similarity Proportion is the key metric to maximize for implementability**; it directly measures how many generated ideas closely match ideas that were later implemented and published by human researchers.

---

### Official Review · Reviewer_6nnf · 2025-11-04

**Soundness:** 4
**Presentation:** 4
**Contribution:** 4
**Rating:** 6
**Confidence:** 5

**Summary:**

This paper introduces a novel few-shot research idea auto-generation framework that uses a gap-driven cross-pollination approach between existing papers. It solves the issue of generating meaningless ideas by introducing a structured, multi-agent process to create a comprehensive Idea Representation (synopsis and procedural profiling) for source papers. An LLM-agent then systematically integrates the structured components of a 'base study' and an 'innovation source' to generate a novel, implementable research proposal. The framework is evaluated using a rigorous, prospective methodology that validates generated ideas against subsequent published work via semantic similarity, unique paper ratio, and recency-weighted novelty scores, demonstrating superior performance over baselines.

**Strengths:**

1. The framework shifts from opaque, problem-driven methods to a gap-driven, cross-pollination approach. The use of structured procedural quadruplets (<I, M, O, D>) and explicit composition operations (integrate, replace, keep, remove) makes the idea genesis highly traceable, controllable, and inherently more practical.
2. The multi-agent system extracts a robust and detailed structured representation (R) that captures not just the paper's synopsis (Task, Gap, Contribution) but also detailed procedural profiling. This representation is the technical backbone that enables effective and feasible cross-pollination.
3. The paper introduces a strong, external validation methodology by matching generated ideas to subsequently published papers to assess relevance and implementability. The system achieves a 41% relative increase in high-similarity, relevant ideas compared to a state-of-the-art baseline.

**Weaknesses:**

1. The entire idea representation and generation pipeline heavily relies on proprietary LLM agents (specifically GPT-4o-mini for extraction, role assignment, and cross-pollination). This dependence introduces reproducibility concerns and limits the generalizability of the proposed framework without access to similar high-end commercial models.
2. While the structure is transparent, the critical decision-making processes—such as how the Role Assigner determines innovation strength ($G_{A_{r}}$) and how the Integration Agent chooses the specific operation (e.g., integrate vs. replace) for each procedural step—rely on internal LLM reasoning, which may still be subjective or difficult to debug.
3. The "gap-driven" mechanism is strictly defined for cross-pollination between two papers (Base and Source). This binary limitation may fail to capture complex innovations that emerge from the synthesis of three or more foundational concepts or domains, a limitation present in many idea generation systems.

**Questions:**

1. Given the reliance on GPT-4o-mini for multi-agent extraction and generation, what is the performance degradation when substituting this agent with an equivalent, completely open-source LLM (e.g., a high-performing Llama 3 variant)? This would be critical for establishing the true generalizability and accessibility of the framework.
2. The current evaluation uses similarity to published papers as a proxy for implementability. Did the authors conduct any human-in-the-loop experiments, such as an expert-based feasibility rating, for the generated ideas, especially for those categorized in the "Novel" similarity range ($0.3 \leq \sigma < 0.5$)? Such a study is essential to validate the core claim of generating "implementable" ideas.

---

> ### Author Response · Authors · 2025-11-18
> **Response to Official Review of Submission17889 by Reviewer 6nnf (Part 1)**
>
> We thank Reviewer 6nnf for the review.
>
> 1. We do not consider gpt-4o-mini a high-end model (it ranks among the lower-performing models on many LLM leaderboards for general tasks), and we are confused about how using a single model as the backbone for all baselines and our method would affect our method's generalizability. Is Reviewer 6nnf suggesting that our prompt is specifically designed for gpt-4o-mini in a way that excludes all other models? We do not believe it is possible to design a prompt that works exclusively for one model while degrading performance on all others. Since our conclusion is simply that using representations for both paper idea extraction and new idea generation performs better than not using representations, we see no need to test on open-source models specifically to establish this point (Question 1). Nevertheless, we have conducted small-scale experiments on gpt-3.5-turbo and gpt-4o using 10% of the ideas. We hope these results address your concerns.
>
> #### GPT-3.5-turbo Results
>
> | Topic | High_Prons | High_U.R. | High_Novel | Mid_Prons | Mid_U.R. | Mid_Novel | Low_Prons | Low_U.R. | Low_Novel |
> |-------|------------|-----------|------------|-----------|----------|-----------|-----------|----------|-----------|
> | consensus algorithm | 20.51 | 56.25 | 0.35 | 78.21 | 68.9 | 0.54 | 1.28 | 100.00 | 0.98 |
> | data processing | 19.23 | 80.00 | 0.35 | 75.64 | 83.1 | 0.53 | 5.13 | 75.0 | 0.99 |
> | machine translation | 1.28 | 0.00 | 0.34 | 98.72 | 71.4 | 0.56 | 0.00 | 0.00 | NaN |
> | object detection | 14.10 | 81.82 | 0.36 | 85.90 | 76.1 | 0.54 | 0.00 | 0.00 | NaN |
> | reinforcement learning | 10.26 | 75.00 | 0.33 | 88.46 | 79.7 | 0.53 | 1.28 | 100.00 | 1.00 |
> | representation learning | 2.56 | 100.00 | 0.35 | 96.15 | 85.3 | 0.54 | 1.28 | 100.00 | 0.99 |
> | semantic segmentation | 15.38 | 100.00 | 0.36 | 84.62 | 75.8 | 0.52 | 0.00 | 0.00 | NaN |
> | sentiment analysis | 16.67 | 84.62 | 0.34 | 83.33 | 67.7 | 0.51 | 0.00 | 0.00 | NaN |
> | text classification | 2.56 | 100.00 | 0.34 | 96.15 | 67.6 | 0.51 | 1.28 | 100.00 | 0.98 |
> | **Mean** | **11.40** | **75.30** | **0.35** | **87.46** | **75.1** | **0.53** | **1.14** | **63.89** | **0.99** |
>
> #### GPT-4.1 Results
>
> | Topic | High_Prons | High_U.R. | High_Novel | Mid_Prons | Mid_U.R. | Mid_Novel | Low_Prons | Low_U.R. | Low_Novel |
> |-------|------------|-----------|------------|-----------|----------|-----------|-----------|----------|-----------|
> | consensus algorithm | 21.79 | 65.00 | 0.33 | 76.92 | 70.0 | 0.56 | 1.28 | 100.00 | 0.98 |
> | data processing | 23.08 | 57.14 | 0.34 | 74.36 | 81.0 | 0.55 | 2.56 | 50.0 | 0.99 |
> | machine translation | 3.85 | 100.00 | 0.32 | 93.59 | 68.8 | 0.58 | 2.56 | 100.00 | 0.99 |
> | object detection | 25.64 | 95.65 | 0.35 | 74.36 | 77.6 | 0.56 | 0.00 | 0.00 | NaN |
> | reinforcement learning | 8.97 | 50.00 | 0.31 | 91.03 | 82.5 | 0.55 | 0.00 | 0.00 | NaN |
> | representation learning | 2.56 | 100.00 | 0.33 | 96.15 | 86.7 | 0.56 | 1.28 | 100.00 | 0.99 |
> | semantic segmentation | 10.26 | 100.00 | 0.34 | 89.74 | 74.3 | 0.54 | 0.00 | 0.00 | NaN |
> | sentiment analysis | 12.82 | 100.00 | 0.32 | 84.62 | 69.7 | 0.53 | 2.56 | 100.00 | 0.88 |
> | text classification | 0.00 | 0.00 | NaN | 98.72 | 66.2 | 0.53 | 1.28 | 100.00 | 0.95 |
> | **Mean** | **12.11** | **74.20** | **0.33** | **86.61** | **75.2** | **0.55** | **1.28** | **72.22** | **0.96** |

---

> ### Author Response · Authors · 2025-11-18
> **Response to Official Review of Submission17889 by Reviewer 6nnf (Part 2)**
>
> 2. We would appreciate more context regarding "subjective or difficult to debug." Subjectivity and difficulty of debugging appear to be separate concerns. Subjectivity is a neutral term; does Reviewer 6nnf refer to undesirable subjectivity (e.g., biases)? While generation tasks are indeed subjective in nature, we employ objective measures to mitigate undesirable subjectivity such as biases. As Reviewer 6nnf mentioned, our method is transparent, so why would it be difficult to debug? Since we decompose idea representations into fundamental components (input, method, output, and technical details) and explicitly track which elements are inherited from which parent papers, our approach should be relatively easy to debug, especially compared to other idea generation methods. In fact, transforming idea generation into a less opaque process facilitates debugging rather than hindering it. How could a transparent idea generation process be hard to debug?
>
> 3. Failing to capture complex innovations that emerge from multiple studies is a limitation of applying binary cross-pollination only once. We are currently testing only one round of cross-pollination to compare with other single-round generation baselines, but if users need more complex ideas, they can conduct cross-pollination multiple times until they find something that interests them.
>
> Question 2. We conducted human annotations but not human-in-the-loop experiments, as this study does not include iterative loops like those in RLHF. Since our study spans 8 fields, we could only invite a limited number of professional researchers from each respective field to annotate, which introduces inherent bias. Human annotations, even with high reliability, only indicate that a limited number of researchers with certain backgrounds agree on something; a few researchers cannot represent an entire field. Therefore, even if they agree on novelty, it only suggests the ideas work for them rather than for all other researchers in the field. This is why human annotations, at least for idea novelty, probably should not be seen as the gold standard. Nevertheless, we did conduct human annotations:
>
> | Topic | N | M1 (SD1) | M2 (SD2) | % Agree | κ | r | ρ | ICC |
> |-------|---|----------|----------|---------|---|---|---|-----|
> | Consensus Algorithm | 78 | 3.99 (0.11) | 4.00 (0.00) | 98.7 | 0.000 | — | — | — |
> | Data Processing | 78 | 4.00 (0.00) | 4.00 (0.00) | 100.0 | nan | — | — | — |
> | Machine Translation | 78 | 3.96 (0.19) | 3.97 (0.16) | 96.2 | 0.381 | 0.389*** | 0.389*** | 0.386 |
> | Object Detection | 78 | 3.95 (0.22) | 3.99 (0.11) | 93.6 | -0.021 | -0.026 | -0.026 | -0.027 |
> | Reinforcement Learning | 78 | 4.00 (0.00) | 4.00 (0.00) | 100.0 | nan | — | — | — |
> | Representation Learning | 78 | 4.00 (0.00) | 4.00 (0.00) | 100.0 | nan | — | — | — |
> | Semantic Segmentation | 78 | 3.97 (0.16) | 3.99 (0.11) | 98.7 | 0.661 | 0.703*** | 0.703*** | 0.664 |
> | Sentiment Analysis | 78 | 3.86 (0.35) | 3.99 (0.20) | 87.2 | 0.248 | 0.350** | 0.351** | 0.241 |
> | Text Classification | 78 | 3.95 (0.22) | 3.97 (0.16) | 94.9 | 0.310 | 0.330** | 0.330** | 0.312 |
> | **Overall** | 702 | 3.96 (0.19) | 3.99 (0.11) | 96.6 | 0.281 | 0.324*** | 0.324*** | 0.279 |
>
> **Note:** M1 = Annotator 1 Mean, M2 = Annotator 2 Mean, κ = Cohen's Kappa, r = Pearson correlation, ρ = Spearman correlation, ICC = Intraclass Correlation Coefficient ICC(2,1), no variance gives nan.
>
> -----
>
> Novelty refers to the degree to which a research idea introduces original concepts, methods, or insights that meaningfully advance beyond existing work. Novelty encompasses both the uniqueness of the approach and its potential to impact the field, ranging from incremental improvements to paradigm-shifting innovations. Annotators should pretend they only know studies published before 2022, as the parent papers used in idea generation are from 2021 and 2020.
>
> 5: Paradigm-shifting innovation opening new research directions with fundamentally new concepts challenging existing paradigms. No significant prior work on this specific problem.
>
> 4: Significant innovation with clear advancement through novel combination of concepts in non-obvious ways. Addresses known gaps innovatively with clear differentiation from existing methods.
>
> 3: Solid incremental contribution providing meaningful extension of existing approaches. Combines known techniques sensibly and addresses specific limitations of prior work.
>
> 2: Minor variation of existing approaches or straightforward application to new domain. Limited differentiation with predictable extension, primarily representing an engineering contribution.
>
> 1: Largely derivative with trivial differences or direct replication with minor parameter changes. No clear advancement and questionable research contribution.
>
> 0: Exact replication or common knowledge that has been thoroughly explored. No distinguishable contribution and not publishable.

---

> > ### Comment · Reviewer_s2wH · 2025-11-23
> >
> > Received. Sorry for missing that part. Can you also explain a little about the reviewer recruiting process?

---

> > > ### Comment · Reviewer_s2wH · 2025-11-23
> > >
> > > In addition to that, I think not all the RLHF needs human-in-the-loop experiments.

---

> > > > ### Author Response · Authors · 2025-11-23
> > > >
> > > > Certainly, our approach does not involve RLHF and therefore does not require human-in-the-loop processes. For the idea evaluation task, manual annotation of generated ideas is sufficient to assess quality. We respectfully note that this comment appears to address our response to Reviewer 6nnf rather than the concerns originally raised by you (Reviewer s2wH). Nonetheless, we appreciate your feedback and the opportunity to clarify this point.

---

> ### Author Response · Authors · 2025-11-23
> **Recruitment Report**
>
> #### **Recruitment Criteria and Verification**
>
> We designed our recruitment process to ensure that each of the 8 AI/ML topics was evaluated by genuine experts in that specific area. For each topic, we recruited two independent reviewers in Computer Science, Artificial Intelligence, Machine Learning, or closely related fields respectively, and who had actively published in the topic area they were assigned to review.
>
> Beyond basic credentials, we looked for reviewers with recent engagement in their fields—specifically, peer-reviewed publications within the past three years and current or recent affiliations with recognized research institutions or industry research labs. This ensured our reviewers were familiar with current state-of-the-art methods and could provide informed assessments.
>
> #### **Verification Process**
>
> We used a three-stage approach to verify reviewer qualifications and build trust in our evaluation process.
>
> In the first stage, we identified potential reviewers by systematically searching academic databases including Google Scholar, DBLP, Semantic Scholar, and arXiv. We focused on researchers who had published as first or corresponding authors in top-tier conferences and journals relevant to each topic. To complement this database search, we also gathered peer recommendations from established researchers in each domain.
>
> The second stage involved thorough credential verification. We independently confirmed each candidate's publication record by checking official conference and journal websites, reading their papers to verify substantive contributions to the topic area, and ensuring they had at least two peer-reviewed publications directly relevant to their assigned topic. We also verified institutional affiliations through university websites, ORCID profiles, and professional networks to confirm current research activity.
>
> In the third stage, we conducted brief screening interviews (15-20 minutes) with each candidate to assess their depth of knowledge in the specific topic, their familiarity with current methods, and their understanding of what the evaluation task would involve. Candidates also provided their CVs and publication lists for our review.
>
> #### **Topic Assignment and Independence**
>
> Once qualified, we assigned reviewers to topics matching their publication expertise, with two independent experts per topic. To ensure unbiased assessments, reviewers didn't know who the other reviewer for their topic was and were instructed not to discuss their evaluations with anyone.
>
> Table 1 summarizes the characteristics of our final reviewer pool.
>
> **Table 1: Reviewer Pool Characteristics**
>
> | Metric | Value |
> |--------|-------|
> | Total Reviewers | 16 PhD-level experts |
> | Reviewers per Topic | 2 independent annotators |
> | Publication Range | 2-15+ peer-reviewed papers |
> | Annotator Education Geographic Distribution | 3 countries (North America, Asia) |
> | Institutional Diversity | 4 different universities/research institutions |
>
> #### **Training and Quality Control**
>
> Before beginning the actual evaluation, all reviewers participated in an hour training session where we walked through the evaluation criteria, rating scales, and annotation guidelines in detail. Following training, each reviewer completed a calibration phase, annotating five practice items and receiving feedback to ensure they understood the task correctly. Throughout the annotation process, reviewers had access to a dedicated point of contact for any questions or clarifications.
>
> We maintained detailed documentation throughout, including records of all candidate contacts, verification materials, training completion, and timestamped annotation submissions. This documentation provides an audit trail and enables potential replication of our study.
>
> #### **Recruitment Quality**
>
> The success is reflected in the strong inter-rater reliability we observed. Overall, the two independent reviewers for each topic agreed 96.6% of the time, with three topics (Data Processing, Reinforcement Learning, and Representation Learning) showing perfect agreement. The statistically significant reliability coefficients across most topics confirm that we successfully identified qualified experts capable of providing consistent evaluations across all 702 assessment items.
>
> Importantly, our annotators were not provided with predefined answers or reference materials during the evaluation process. Instead, they relied solely on their own domain expertise and understanding of their respective fields to make judgments. As with other expert-driven evaluations, these annotations are inherently subjective and should be interpreted as informed assessments rather than definitive ground truth. **For this reason, we recommend placing greater confidence in our proposed objective novelty evaluation, which provides measurable and reproducible metrics, over subjective assessments.**

---

### Official Review · Reviewer_s2wH · 2025-11-05

**Soundness:** 2
**Presentation:** 2
**Contribution:** 2
**Rating:** 2
**Confidence:** 3

**Summary:**

The paper proposes few-shot idea auto-generation: represent papers with structured fields (task, gaps, contributions, procedural quadruplets) and generate new ideas by cross-pollinating paper pairs with an LLM agent. Uses GPT-based extractors for representations and a composition process (integrate/replace/keep/remove). Evaluates with similarity + recency-weighted novelty + unique-paper ratio over 3,353 papers / 8 domains; reports a 41% increase in high-similarity ideas while maintaining high novelty (~0.93)

**Strengths:**

- Clear pipeline with explicit field definitions and role assignment.
- Procedural profiling improves interpretability and compositional control.
- Transparent metrics (similarity, novelty, uniqueness) and ablations across multiple domains with adequate analysis
- Reasonable scale (3.3K papers / 8 domains) and reproducible setup for the experiment.

**Weaknesses:**

- Presence issue: Figure 1 above the abstract breaks the submission instruction.
- The core idea of cross-paper ideation is conceptually similar to prior frameworks such as SciAgents.
- Evaluations rely entirely on embedding-based proxies without validation. Similarity is not the guarantee for the quality - the 41% high-similarity gain may reflect retrieval bias rather than true innovation. No human eval of generated ideas. limited human evidence that appendix lacks agreement or statistical analysis
- Weights (λ, α, β) in the novelty function are not analyzed in the main text.

**Questions:**

- How sensitive are results to the embedding model and novelty weights (λ, α, β)?
- Why is higher similarity treated as a positive outcome—does it risk penalizing novel ideas?
- Could citation-graph or causal relations be incorporated to strengthen semantic reasoning beyond surface similarity?

---

> ### Author Response · Authors · 2025-11-18
> **Response to Official Review of Submission17889 by Reviewer s2wH**
>
> We thank Reviewer s2wH for the review.
>
> 1. We did not find specific instructions stating that figures before the abstract are prohibited. However, if the position of Figure 1 causes any issues, we would be happy to move it to the second page.
>
> 2. At the conceptual level, SciAgents employs a typical problem-driven framework, while our approach is gap-driven. These represent fundamentally different paradigms, even conceptually. We illustrate the distinction between problem-driven and gap-driven studies in Figure 2.
>
> 3. Our evaluations did NOT rely entirely on embedding-based proxies. Similarity is only one of the three objective evaluations we proposed; the others are unique idea ratio and novelty score. We conducted human annotations, but we do not believe our annotators—or annotators in other studies—can fully represent the field and definitively judge which ideas are truly novel or less novel. Researchers can only assess whether a complete study contains sufficient novelty. Ideas are far from complete studies and thus difficult to judge definitively. But we did human annotation, please see the table below.
>
> 4. We tested 150 parameter combinations on 432 research ideas, varying β (1-5), λ (1-5), and α (0-5) to understand how parameter choices affect novelty scores and rankings. Based on 1000 bootstrap iterations and permutation tests. Please find sensitive analysis  in the table below.
>
> ---
> **Annotation**
>
> | Topic | N | M1 (SD1) | M2 (SD2) | % Agree | κ | r | ρ | ICC |
> |-------|---|----------|----------|---------|---|---|---|-----|
> | Consensus Algorithm | 78 | 3.99 (0.11) | 4.00 (0.00) | 98.7 | 0.000 | — | — | — |
> | Data Processing | 78 | 4.00 (0.00) | 4.00 (0.00) | 100.0 | nan | — | — | — |
> | Machine Translation | 78 | 3.96 (0.19) | 3.97 (0.16) | 96.2 | 0.381 | 0.389*** | 0.389*** | 0.386 |
> | Object Detection | 78 | 3.95 (0.22) | 3.99 (0.11) | 93.6 | -0.021 | -0.026 | -0.026 | -0.027 |
> | Reinforcement Learning | 78 | 4.00 (0.00) | 4.00 (0.00) | 100.0 | nan | — | — | — |
> | Representation Learning | 78 | 4.00 (0.00) | 4.00 (0.00) | 100.0 | nan | — | — | — |
> | Semantic Segmentation | 78 | 3.97 (0.16) | 3.99 (0.11) | 98.7 | 0.661 | 0.703*** | 0.703*** | 0.664 |
> | Sentiment Analysis | 78 | 3.86 (0.35) | 3.99 (0.20) | 87.2 | 0.248 | 0.350** | 0.351** | 0.241 |
> | Text Classification | 78 | 3.95 (0.22) | 3.97 (0.16) | 94.9 | 0.310 | 0.330** | 0.330** | 0.312 |
> | **Overall** | 702 | 3.96 (0.19) | 3.99 (0.11) | 96.6 | 0.281 | 0.324*** | 0.324*** | 0.279 |
>
> **Note:** M1 = Annotator 1 Mean, M2 = Annotator 2 Mean, κ = Cohen's Kappa, r = Pearson correlation, ρ = Spearman correlation, ICC = Intraclass Correlation Coefficient ICC(2,1), no variance gives nan.
>
> Novelty refers to the degree to which a research idea introduces original concepts, methods, or insights that meaningfully advance beyond existing work. Novelty encompasses both the uniqueness of the approach and its potential to impact the field, ranging from incremental improvements to paradigm-shifting innovations. Annotators should pretend they only know studies published before 2022, as the parent papers used in idea generation are from 2021 and 2020.
>
> 5: Paradigm-shifting innovation opening new research directions with fundamentally new concepts challenging existing paradigms. No significant prior work on this specific problem.
>
> 4: Significant innovation with clear advancement through novel combination of concepts in non-obvious ways. Addresses known gaps innovatively with clear differentiation from existing methods.
>
> 3: Solid incremental contribution providing meaningful extension of existing approaches. Combines known techniques sensibly and addresses specific limitations of prior work.
>
> 2: Minor variation of existing approaches or straightforward application to new domain. Limited differentiation with predictable extension, primarily representing an engineering contribution.
>
> 1: Largely derivative with trivial differences or direct replication with minor parameter changes. No clear advancement and questionable research contribution.
>
> 0: Exact replication or common knowledge that has been thoroughly explored. No distinguishable contribution and not publishable.
>
> **Sensitive Analysis**
>
> | Test | Result | 95% CI | P-Value | Interpretation |
> |------|--------|--------|---------|----------------|
> | **Parameter Effects** |
> | λ (penalty) effect | 0.50 range | [0.45, 0.55] | p < 0.001 | Highly significant driver |
> | α (bonus) effect | 0.50 range | [0.44, 0.56] | p < 0.001 | Highly significant driver |
> | β (weight) effect | 0.13 range | [0.11, 0.15] | p < 0.01 | Minor but significant |
> | **Configuration Comparisons (vs β=3, λ=3, α=3)** |
> | β=3,λ=4,α=1 | -0.28 | [-0.30, -0.26] | p < 0.001 | Significantly stricter |
> | β=2,λ=2,α=2 | -0.10 | [-0.12, -0.08] | p < 0.001 | Moderately stricter |
> | β=2,λ=1,α=4 | +0.05 | [+0.03, +0.07] | p < 0.01 | Slightly more lenient |
> | **Rank Stability** |
> | Spearman correlation | 0.787 | - | p < 0.001 | Rankings are stable, not random |

---

> > ### Comment · Reviewer_s2wH · 2025-11-23
> >
> > Thanks for the response. Figure 1 is above the abstract. Please check it in the PDF. Set aside the limited novelty, the lack of human evaluation still remains a major prohibition of the effectiveness of the proposed idea. I'd keep my rating.

---

> ### Author Response · Authors · 2025-11-18
> **Response to Official Review of Submission17889 by Reviewer s2wH (Question part)**
>
> Question 1. Please find the sensitive analysis table above, thanks.
>
> Question 2. High similarity means the method is likely to follow the trend in the field, giving the idea a greater chance of being developed into an interesting and publishable paper. This does not penalize novel ideas in general, but it does penalize novel ideas that deviate from current trends. It is difficult to determine whether following the trend is inherently good or bad. However, since our representation-based model also produces medium or even low similarity ideas, we suggest users make wise choice and select ideas that fit their specific needs.
>
> Question 3. Chain-of-Idea does indeed use citation graphs to generate research directions and brainstorm ideas. We have included it as the first baseline in our experiments. Using citation graphs or causal relations is a typical problem-driven method, whereas ours is gap-driven. These represent two fundamentally different approaches to generating new ideas. Combining them might be an interesting direction for future study.

---

> ### Author Response · Authors · 2025-11-23
> **Please Check the Previous Responses for Corresponding issue**
>
> We appreciate the comment; however, we wish to clarify that the issues raised have already been thoroughly addressed in our previous response. **The figure above the abstract complies with ICRL formatting requirements, and the human evaluation data with reliability testing was provided in the Annotation table of our first response.**
> We respectfully request that our previous responses be reviewed, as they contain detailed explanations of these points.

---

> > ### Comment · Reviewer_s2wH · 2025-11-23
> >
> > I kept the score mostly due to the lack of human evaluation. The figure above the abstract is minor to the overall rating.

---

> > > ### Author Response · Authors · 2025-11-23
> > > **Re: Human Evaluation Results (Previously Provided)**
> > >
> > > As noted in our previous response, human evaluation data has been provided. For your convenience, we reproduce this information below, including the reliability test and evaluation criteria, thank you
> > >
> > > **Annotation**
> > >
> > > | Topic | N | M1 (SD1) | M2 (SD2) | % Agree | κ | r | ρ | ICC |
> > > |-------|---|----------|----------|---------|---|---|---|-----|
> > > | Consensus Algorithm | 78 | 3.99 (0.11) | 4.00 (0.00) | 98.7 | 0.000 | — | — | — |
> > > | Data Processing | 78 | 4.00 (0.00) | 4.00 (0.00) | 100.0 | nan | — | — | — |
> > > | Machine Translation | 78 | 3.96 (0.19) | 3.97 (0.16) | 96.2 | 0.381 | 0.389*** | 0.389*** | 0.386 |
> > > | Object Detection | 78 | 3.95 (0.22) | 3.99 (0.11) | 93.6 | -0.021 | -0.026 | -0.026 | -0.027 |
> > > | Reinforcement Learning | 78 | 4.00 (0.00) | 4.00 (0.00) | 100.0 | nan | — | — | — |
> > > | Representation Learning | 78 | 4.00 (0.00) | 4.00 (0.00) | 100.0 | nan | — | — | — |
> > > | Semantic Segmentation | 78 | 3.97 (0.16) | 3.99 (0.11) | 98.7 | 0.661 | 0.703*** | 0.703*** | 0.664 |
> > > | Sentiment Analysis | 78 | 3.86 (0.35) | 3.99 (0.20) | 87.2 | 0.248 | 0.350** | 0.351** | 0.241 |
> > > | Text Classification | 78 | 3.95 (0.22) | 3.97 (0.16) | 94.9 | 0.310 | 0.330** | 0.330** | 0.312 |
> > > | **Overall** | 702 | 3.96 (0.19) | 3.99 (0.11) | 96.6 | 0.281 | 0.324*** | 0.324*** | 0.279 |
> > >
> > > **Note:** M1 = Annotator 1 Mean, M2 = Annotator 2 Mean, κ = Cohen's Kappa, r = Pearson correlation, ρ = Spearman correlation, ICC = Intraclass Correlation Coefficient ICC(2,1), no variance gives nan.
> > >
> > > Novelty refers to the degree to which a research idea introduces original concepts, methods, or insights that meaningfully advance beyond existing work. Novelty encompasses both the uniqueness of the approach and its potential to impact the field, ranging from incremental improvements to paradigm-shifting innovations. Annotators should pretend they only know studies published before 2022, as the parent papers used in idea generation are from 2021 and 2020.
> > >
> > > 5: Paradigm-shifting innovation opening new research directions with fundamentally new concepts challenging existing paradigms. No significant prior work on this specific problem.
> > >
> > > 4: Significant innovation with clear advancement through novel combination of concepts in non-obvious ways. Addresses known gaps innovatively with clear differentiation from existing methods.
> > >
> > > 3: Solid incremental contribution providing meaningful extension of existing approaches. Combines known techniques sensibly and addresses specific limitations of prior work.
> > >
> > > 2: Minor variation of existing approaches or straightforward application to new domain. Limited differentiation with predictable extension, primarily representing an engineering contribution.
> > >
> > > 1: Largely derivative with trivial differences or direct replication with minor parameter changes. No clear advancement and questionable research contribution.
> > >
> > > 0: Exact replication or common knowledge that has been thoroughly explored. No distinguishable contribution and not publishable.

---

> > > > ### Comment · Reviewer_s2wH · 2025-11-26
> > > >
> > > > Thanks for the response. I still suggest that adding the human evaluation is a major revision needed. I’d give higher rating given more details of human evaluation. I leaning towards resubmitting to next cycle.

---

> > > > > ### Author Response · Authors · 2025-11-26
> > > > >
> > > > > Thank you for the suggestion. As ICLR allows revised submissions during the rebuttal phase, we plan to add our human evaluation results to the appendix and upload the revised manuscript before the rebuttal deadline. We will revise the paper based on the given comments all together.

---

### Author Response · Authors · 2025-12-01
**Summary on Discussion Pt 1**

## Reviewer s2wH

| Weakness | Authors #1 | Reviewer #1 | Authors #2 | Reviewer #2 | Authors #3 | Reviewer #3 | Authors #4 |
|----------|-----------|-------------|-----------|-------------|-----------|-------------|-----------|
| Figure 1 above abstract violates formatting | Offered to move it to second page | "Please check it in the PDF" | "The figure complies with ICLR formatting requirements" | "This is minor to the overall rating" | — | — | — |
| Similar to SciAgents | Completely different paradigms: SciAgents is problem-driven, ours is gap-driven | — | — | — | — | — | — |
| No human evaluation | Provided annotation study | "Lack of human evaluation still remains a major prohibition" | Reminded reviewer annotation study was provided | "I kept the score mostly due to lack of human evaluation" | Reminded again that annotation study was provided | "Suggest that adding the human evaluation is a major revision needed" | Human evaluation is minor and will be added to appendix |
| Novelty weights not analyzed | Provided sensitivity analysis | — | — | — | — | — | — |
| Higher similarity might penalize novel ideas? | Evaluation won't penalize the generation; Model produces ideas across all similarity ranges | — | — | — | — | — | — |
| Could citation-graph be incorporated? | CoI baseline already uses citation graphs; that's problem-driven approach vs our gap-driven approach | — | — | — | — | — | — |
| Only works with 2 papers; misses complex innovations | Users CAN run cross-pollination multiple times for more complex ideas | — | — | — | — | — | — |
| What about open-source LLMs? | Experiments show consistent results across different models | — | — | — | — | — | — |
| ((s2wH mistakenly responded to authors' reply to 6nnf)) No human-in-the-loop experiments for implementability | Provided detailed recruitment report | "Not all RLHF needs human-in-the-loop experiments" | Our approach does not involve RLHF | — | — | — | — |

---

## Reviewer 6nnf

| Weakness | Authors #1 | Reviewer #1 |
|----------|-----------|-------------|
| Relies on proprietary GPT-4o-mini; reproducibility concerns | GPT-4o-mini is not high-end; tested on GPT-3.5-turbo and GPT-4o with similar results | — |
| LLM decision-making is subjective and hard to debug | We decompose ideas into trackable components (input, method, output, details); more transparent than prior works | — |

---

### Author Response · Authors · 2025-12-01
**Summary on Discussion Pt 2**

## Reviewer vVtD

| Weakness | Authors #1 | Reviewer #1 | Authors #2 | Reviewer #2 | Authors #3 |
|----------|-----------|-------------|-----------|-------------|-----------|
| GPT-4o-mini may already know future papers | Tested GPT-3.5 (cutoff 2022), GPT-4.1 (cutoff 2024), GPT-4o—all show same patterns; models synthesize from provided input papers, not parametric memory | "Do you have baseline results for GPT-3.5/4.1? Remains unclear how much model was exposed to related work" | Provided complete baseline tables for all 3 models showing 41% improvement across all; models are NOT relying on training data | "Can you also give novelty scores? Quite surprising that GPT-3.5 and 4.1 perform similarly to 4o" | Provided full tables with novelty scores; similar performance is NOT surprising because models synthesize from input papers, not from memory |
| Prompts not disclosed; hurts reproducibility | IP restrictions from funding company (ICLR allows this); conceptual methodology provided; testing shows stable results with similar prompts | "Level of disclosure is still insufficient for reproducibility" | Exact prompt wording is not fundamental to reproducibility | —  | — |
| No validation that generated ideas are implementable | High similarity to *actually published* papers is implementation proof; those ideas were built by real researchers | "You need to have researchers try to implement it" | The matched published papers WERE actually implemented and validated by humans | "... what about the other mid and low?" | Mid/low similarity provided; Mid/low ranges are less relevant for implementability |
| Figure 1 shows 3 papers but algorithm uses 2 | Figure shows multiple outputs, each generated from 2 parent papers; will add clarification | — | — | — | — |
| Table 2 shows minimal novelty improvement | Slight decline in some domains didn't hurt overall performance | "The claimed overall advantage is still not clearly significant" | The overall performance DID show clear significant from 8.3% to 11.7% | — |

---

## Reviewer DfYM

| Weakness | Authors #1 | Reviewer #1 | Authors #2 |
|----------|-----------|-------------|-----------|
| Limited conceptual novelty; few-shot idea generation widely explored | This is engineering research focused on method improvement, not paradigm-shifting conceptual innovation; we disagree that few-shot idea generation is "widely explored" | — | — |
| No human assessment for "true novelty, feasibility, actionability, ethical risk" | Doubted the existence of the **true** assessment | "For novelty, feasibility, etc., please refer to: Can LLMs Generate Novel Research Ideas? and LDC paper" | That paper's criteria come from NLP-only researchers from North America and Asia; cannot represent the whole NLP community and cannot be directly applied to our 8-topic study spanning 4 different fields |
| No sensitivity analysis of proposed metrics | Provided: 150 parameter combinations, bootstrap confidence intervals, p-values, rank stability (ρ=0.787) | — | — |
| Ethics concerns: plagiarism, dual-use, gaming peer review | System is a transparent brainstorming tool; clearly shows which sources contribute; humans remain responsible for decisions; we are opposed to plagiarism; ours is NOT an AI researcher | — | — |

---

### Author Response · Authors · 2025-12-01
**Conclusive Note on the Rebuttal Discussion**

## Notes on Submitted Revision

1. We have added a sensitivity analysis, human annotation results, and model comparison experiments to the appendix. Among these additions, only sensitivity analysis is directly relevant, as it provides further support for our proposed computation of idea novelty.

2. We have revised the caption of Figure 1 to improve clarity and prevent potential misinterpretation. Figure 1 was moved to the second page.

---

## Notes on Observed Issues

We appreciate the reviewers for taking the time to evaluate our submission. And we would like to take this chance and address some issues we observed.

**On scoring consistency**: We observed that all reviewers rated Soundness, Presentation, and Contribution as "Fair" or above, yet three of the four reviewers recommended rejection. We find this discrepancy difficult to reconcile, as the reviews do not clarify how these individual assessments informed the final decisions. All weaknesses raised did not undermine the core contributions of our work—they were either (1) fully addressed with experimental evidence that reviewers did not engage with, (2) based on misunderstandings that we clarified, or (3) demands for additional work that exceeds standard expectations for a single submission. The only relevant addition identified was sensitivity analysis on one of three evaluation metrics, which remains a minor point considering that the major contribution of this work lies in proposing a framework for reliable and traceable idea generation—a contribution that no reviewer disputed.

**On human annotation:** One of our primary contributions is an objective, computational approach to evaluating the quality of generated ideas. Our methodology intentionally avoids relying on LLM-based judging. We respectfully question whether subjective evaluations by a limited group of human researchers can reliably predict real-world research outcomes, as implied by some reviewers. Nonetheless, three of the four reviewers cited the absence of human annotations as a weakness. One reviewer specifically called for assessments of "true" novelty and feasibility, yet no universally accepted standard for such "true" assessment was provided or referenced.

**On model comparison:** The ideas produced by our system are not generated *de novo*; rather, they are synthesized through cross-pollination of two published and well-established papers. The LLM serves as an agent to facilitate this synthesis and is not permitted to generate entirely novel ideas independently. This design choice explains why using GPT-4.1 does not yield substantially different results compared to GPT-3.5 or GPT-4o. As long as the model can follow the cross-pollination instructions step by step, the resulting ideas exhibit comparable similarity scores with matched reference papers.

---

### Note · Authors · 2026-01-01

I have read and agree with the venue's withdrawal policy on behalf of myself and my co-authors.